**Formation and sink of glyoxal and methylglyoxal in a polluted subtropical environment: observation-based photochemical analysis and impact evaluation**

Zhenhao Ling[1], Qianqian Xie[2], Zhe Wang[3,4]*, Tao Wang[5], Hai Guo[5], Xuemei Wang[2,4]*

[1.] School of Atmospheric Sciences, Guangdong Province Key Laboratory for Climate Change and Natural Disaster Studies, Sun Yat-sen University and Southern Marine Science and Engineering Guangdong Laboratory (Zhuhai), Zhuhai, China

[2.] Institute for Environmental and Climate Research, Jinan University, Guangzhou, China

[3.] Division of Environment and Sustainability, The Hong Kong University of Science and Technology, Hong Kong, China

[4.] Guangdong-Hongkong-Macau Joint Laboratory of Collaborative Innovation for Environmental Quality, Guangzhou, China

[5.] Department of Civil and Environmental Engineering, Hong Kong Polytechnic University, Hong Kong, China

*Correspondence: Zhe Wang (z.wang@ust.hk) and Xuemei Wang (eciwxm@jnu.edu.cn)

## Abstract

The dicarbonyls, glyoxal (Gly) and methylglyoxal (Mgly) have been recognized as important precursors of secondary organic aerosols (SOAs) through the atmospheric heterogeneous process. In this study, field measurement was conducted at a receptor site in the Pearl River Delta (PRD) region in south China, and an observation based photochemical box model was subsequently applied to investigate the production and evolution of Gly and Mgly as well as their contributions to SOA formation. The model was coupled with a detailed gas-phase oxidation mechanism of volatile organic compounds (VOCs) (*i.e.,* MCM v3.2), heterogeneous processes of Gly and Mgly (*i.e.,* reversible partitioning in aqueous phase, irreversible volume reactions and irreversible surface uptake processes), and the gas-particle partitioning of oxidation products. The results suggested that without considering the heterogeneous processes of Gly and Mgly on aerosol surfaces, the model would overpredict the mixing ratios of Gly and Mgly by factors of 3.3 and 3.5 compared to the observed levels. The agreement between observation and simulation improved significantly when the irreversible uptake and the reversible partitioning were incorporated into the model, which in total both contributed ~62% to the destruction of Gly and Mgly during daytime, respectively. Further analysis on the photochemical budget of Gly and Mgly showed that the oxidation of aromatics by the OH radical was the major pathway producing Gly and Mgly, followed by

degradation of alkynes and alkenes. Furthermore, based on the improved model mechanism, the contributions of VOCs oxidation to SOA formed from gas-particle partitioning ($SOA_{gp}$) and from heterogeneous processes of Gly and Mgly ($SOA_{het}$) were also quantified. It was found that *o*-xylene was the most significant contributor to $SOA_{gp}$ formation (~29%), while *m,p*-xylene and toluene made dominant contributions to $SOA_{het}$ formation. Overall, the heterogeneous processes of Gly and Mgly can explain ~21% of SOA mass in the PRD region. The results of this study demonstrated the important roles of heterogeneous processes of Gly and Mgly in SOA formation, and highlighted the need for a better understanding of the evolution of intermediate oxidation products.

**Keywords:** Glyoxal, Methylglyoxal, Secondary organic aerosol, Pearl River Delta, Volatile organic compound, Photochemical box-model

## 1. Introduction

Organic aerosols (OAs) are important components of atmospheric aerosols, with important impacts on radiation balance, air quality, atmospheric oxidative capacity, and climate change (Zhu et al., 2011; Carlton et al., 2009; Hoyle et al., 2009). In addition to the primary organic components (primary OA, POA) directly emitted from various sources in the particulate form, a large fraction of OAs are secondarily produced (SOA) through the aging of POAs, and through complex homogenous/heterogeneous reactions of volatile or semi-volatile organic compounds (VOCs, SVOCs) (Jimenez et al., 2009; Steinfeld and Jeffrey, 1998). SOA has frequently been observed to dominate the OA in many regions, particularly during severe haze pollution events (Guo et al., 2012; Zhang et al., 2017). However, the characteristics of SOAs are still poorly understood because of their complicated formation mechanisms, various chemical compositions, and multitude of precursors from diverse emissions, thus making SOAs an important research topic in the field of the atmospheric environment.

In addition to primary precursors including isoprene, terpene and aromatic hydrocarbons, glyoxal (Gly) and methylglyoxal (Mgly) have been recognized to be of critical importance to SOA formation, especially through heterogeneous and multiphase processes, in many laboratory and model studies (Waxman, et al., 2013, 2015; McNeill

et al., 2015; De Haan et al., 2009; Fu et al., 2008). Many efforts have been made to
investigate the sources, evolution of Gly and Mgly and their contributions to SOA
(Benavent et al., 2019; Zhang et al., 2016; Sumner et al., 2014; DiGangi et al., 2012;
Stavarakou et al., 2009). For example, Li et al (2015) constructed a Master Chemical
Mechanism with an equilibrium partitioning module and coupled it in a Community
Air Quality Model (CMAQ) to predict the regional concentrations of SOA from VOCs
in the eastern United States (U.S). It was found that those SOA formed from Gly and
Mgly were accounted for more than 35% of total SOA. Similarly, Ying et al. (2015)
used a modified SAPRC-11 (S11) photochemical mechanism, considering the surface-
controlled reactive uptake of Gly and Mgly, and incorporated the mechanism in the
CMAQ model to simulate ambient SOA concentrations during summer in the eastern
U.S. The results showed that the uptake of Gly and Mgly resulted in the significant
improvement in predicated SOA concentration, and the aerosol surface uptake of
isoprene-generated Gly, Mgly and epoxydiol accounted for more than 45% of total SOA.
As two smallest dicarbonyl compounds, the sources of Gly and Mgly are
complicated. It has been well documented that Gly and Mgly have limited primary
sources except biomass burning and biofuel combustion (Grosjean et al., 2001; Zhang
et al., 2016). Furthermore, the primary emissions of Gly and Mgly were much less
significant than those secondarily from photochemical reactions (Lv et al., 2019). Fu et
al. (2008) estimated that primary emissions only accounted for about 4% and 17% to
the total emissions of Mgly and Gly, respectively. On a global scale, isoprene and
ethyne are the most important precursors of Gly and Mgly; on the local scale, however,
degradation of aromatics is the major pathway for the production of Gly and Mgly in
urban and sub-urban areas. For example, the oxidation of aromatics contributed to
approximately 75% of Gly formation in Mexico City (Li et al., 2014; Volkamer et al.,

92 2007).

As for the atmospheric sink for Gly and Mgly, photolysis, reaction with OH, dry
deposition, and heterogenous processes are considered as the main loss pathways,
among which aerosol uptake is most complicated and needs more comprehensive
exploration (De Haan et al., 2018; McNeill, 2015; Knote et al., 2014; Fu et al., 2008).
The uptake of Gly and Mgly onto inorganic or organic particles has been studied in
laboratory experiments under controlled conditions (De Haan et al., 2018; Liggio et al.,
2005), and uptake coefficients ($\gamma$) were measured by the loss of gas phase concentration

or the increase of particle organic mass, within the range of $\sim 10^{-4}$ to $10^{-2}$ (De Haan et al., 2018; Pye et al., 2017; Liggio et al., 2005). The lower $\gamma$ value was probably related to the kinetic limitations (Ervens and Volkman 2010), while the higher $\gamma$ value may be associated with the increased particle acidity (Liggio et al., 2005), relative humidity (De Haan et al., 2018; Corrigan et al., 2008) and ionic strength (Kroll et al., 2005). In addition, ammonium-catalyzed and OH reactions were found to have significant influences on the surface uptake of dicarbonyls (Knote et al., 2014; Kampf et al., 2013; Noziere et al., 2008), and the rate coefficients were found to increase with the increasing ammonium ion activity ($a_{NH4+}$) and pH (Noziere et al., 2008). The "salting-in" effects resulted from the increased ionic strength could cause significant increase ($\sim$ 3 orders of magnitude) of henry's law constant for Gly, affecting the gas-aqueous partitioning of Gly and enhanced the available Gly for aqueous reactions (Kampf et al., 2013; Knote et al., 2014; Waxman et al., 2015).

The uptake processes of Gly and Mgly derived from the laboratory studies were incorporated into different models to investigate their formation and destruction (Ge et al., 2011; Knote et al., 2014; Pye et al., 2017). It was found that solely incorporating the irreversible uptake pathways of dicarbonyls could lead to high discrepancy between the observation and simulation results from the global 3D model and other models (Hu et al ., 2017; Li X et al., 2014; Li et al., 2013a), highlighting the needs to consider more comprehensive processes including both reversible and irreversible pathways for better simulating the dicarbonyls. Those previous studies showed that the contribution of heterogeneous processes to the destruction of dicarbonyls varied in the range of 0~80%, which depended on the relative humidity, the precursors incorporated into the model as well as the aerosol concentrations for the given region (Knote et al., 2014).

The Pearl River Delta (PRD) region has been experiencing rapid industrialization and urbanization in the last three decades, making it one of the most developed regions in China. The filed measurement results suggested that OA contributed 30~40% to $PM_{2.5}$ mass, and SOA dominated the OA with fractions up to ~80% in PRD (Huang et al., 2014; He et al., 2011). Furthermore, the contribution of SOA in $PM_{2.5}$ has been increasing in recent years, highlighting the necessity for better understanding the formation of SOA in this region (Wu et al., 2019; Wang et al., 2019). However, model simulation which provides robust information of the influence of physical processes

and chemical degradation in SOA formation still underpredict the SOA abundance with
only traditional VOC precursors incorporated, hindering the better understanding the
sources and formation mechanism of SOA in PRD (Wu et al., 2019; Fu et al., 2012;
Wang et al., 2009). It was found that incorporating emissions of Gly and Mgly, and their
degradation mechanisms could effectively narrow the gap between the measured and
modelled SOA (Fu et al., 2012; Li et al., 2013a). However, only the simple
parameterization of surface uptake of Gly and Mgly without detailed physical and
chemical processes (e.g., reversible partitioning of Gly and Mgly into deliquesced
droplets) in the model could bias the evolution of Gly and Mgly, leading to the poor
understanding on the budgets of Gly and Mgly, their relationship with precursors, and
the contributions of precursors to SOA formation in PRD (De Haan et al., 2018;
Waxman et al., 2015; Knote et al., 2014; Li et al., 2013a, 2014; Lu et al., 2013).
Therefore, to improve the model performance for the simulation of Gly and Mgly and
to investigate their evolution and contribution to SOA formation, the observation data
from a receptor site in the PRD region was analyzed by a photochemical box model
with near-explicit chemical mechanisms (*i.e.,* the master chemical mechanism, MCM),
and improvements with reversible and irreversible heterogeneous processes of Gly and
Mgly, and the gas-particle partitioning of oxidation products in the present study. The
production and evolution of Gly, Mgly, and other intermediate products were
investigated. The observed and simulated levels of Gly and Mgly were compared to
evaluate the performance of the model, which was further used to quantify the
contributions of individual VOCs to SOA formation at the receptor site of PRD.

**2. Methodology**
*2.1. Field measurement*
Field measurements were carried out at Guangdong provincial atmospheric
supersite located at Heshan (22.728°N, 112.929°E, 60 m above sea level) in Jiangmen
City of the PRD region. The sampling site is located about 50 km and 80 km southwest
from Foshan and Guangzhou City, respectively. The Heshan site is surrounded by
mountain areas with trees and subtropical plants, and the location of the site is showed
in Figure 1. Ambient measurement of VOCs, carbonyls and other trace gases was
conducted during January 02- 08, 2017, when the dominant wind was mainly from the
southeast where the center of PRD (*i.e.,* Zhuhai and Zhongshan) was located. A detailed
description of the Heshan site and the measurement methodology was provided in our
previous studies (Chang et al., 2019; Yun et al., 2018).

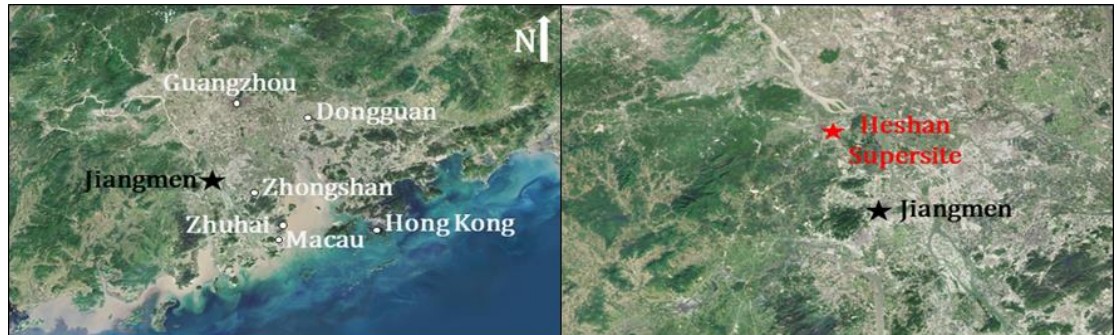


Figure. 1 The location of sampling site and its surrounding environment in the Pearl River Delta
region. The base map was from © Google Maps.

Briefly, mono-carbonyls, Gly and Mgly were collected with 2,4-
dinitrophenylhydrazine cartridges every 3 h and detected using a high-performance
liquid chromatography (HPLC) system (PerkinElmer 200 Series, US). The hourly
VOCs were measured using a cryogen-free automatic gas chromatograph system
equipped with a mass spectrometer and a flame ionization detector (GC-MS/FID)
(Wang et al., 2014). CO, $SO_2$, and $O_3$ was measured using a gas filter correlation
analyzer, a pulsed fluorescence analyzer, and a UV photometric analyzer, respectively
(Thermo Scientific 48i, 43i, 49i). NO and $NO_2$ were detected using a
chemiluminescence instrument (Thermo Scientific 42i) with a photolytic converter (Xu
et al., 2013). The method detection limits for non-methane hydrocarbons (NMHCs),
carbonyls, CO, $SO_2$, $O_3$, NO, and $NO_2$ were 20-300, 20-450, 4000, 100, 500, 60, and
300 pptv, respectively (Yun et al., 2018; Chang et al., 2019; Li et al., 2020). The
measurement method uncertainty of the retrieved Gly and Mgly mixing ratios was
estimated around 15% (Chang et al., 2019; Li et al., 2020). Furthermore, hourly
meteorological parameters including temperature, wind speed, wind direction, pressure,
and relative humidity were recorded using a pyranometer (CMP22, Kipp & Zonen B.V.,
Holland) and a portable weather station (Model WXT520, Vaisala, Finland).

### *2.2. Photochemical box model with master chemical mechanism (PBM-MCM) and*
### *gas-particle partitioning SOA scheme*
Photochemical box model (PBM) was employed in this study to simulate the
oxidation of Gly, Mgly and different VOCs, based on a master chemical mechanism
(MCM) coupled with a gas-particle partitioning module to represent the SOA formation
scheme. The MCM (version 3.2) is a near-explicit mechanism including ~16,500
reactions involving ~6,000 chemical species with the latest IUPAC inorganic
nomenclature, which described the chemical degradation of ~143 primary VOCs and
their oxidation products. The MCM scheme has been applied to different photochemical
box models to investigate the oxidation and reactivates of various VOCs, the formation
of photochemical $O_3$ and secondary organic products, atmospheric radical budget and
propagation, as well as the policy evaluation on mitigating the photochemical smog
(*e.g.,* Ling et al., 2014, 2019; Wang et al., 2017; Lyu et al., 2015; Xue et al., 2014a, b).
The physical processes including dry deposition and atmospheric dilution due to the
variations of planetary boundary layer heights (configured according to the local
observation in the PRD region from previous studies (Li et al., 2014; Wang et al., 2013;
Fan et al., 2011) were considered in the model. Similar to other box models in
simulating the degradation of VOCs and formation of SOA (Aumont, et al., 2012; Lee-
Taylor, et al., 2011; Zhang and Seinfeld, 2013), the PBM-MCM model was developed
by assuming a well-mixed box without consideration of vertical and horizontal
transport, and air pollutants were assumed to be homogeneous (Lam et al., 2013; Ling
et al., 2014). Thus, the influence of horizontal and vertical transport on air pollutants
was not considered in this study.
In addition to the gas-phase degradation of VOCs, a gas-particle partitioning
module for the oxidation products of VOCs and those compounds with an estimated
normal boiling temperature greater than 450 K, as developed by Johnson et al (2006),
were incorporated into the model to represent the SOA formation scheme (Johnson et
al., 2005, 2006; Kamens et al., 1999; Stein et al., 1994). In brief, the gas-to-particle
equilibrium partitioning of the species was described by the partitioning coefficient ($K_p$,
unit: $m^3 \mu g^{-1}$) using Eq. 1 (Johnson et al., 2006).

$$K_p = \frac{7.501 \times 10^{-9} RT}{MW_{Om}\xi P_L^0}$$ (Eq.1)

where $R$, $T$, $MW_{om}$, and $\xi$ are the ideal gas constant (8.314 J K$^{-1}$mol$^{-1}$), temperature (K), the mean molecular weight of the absorbing particle organic matter (g mol$^{-1}$), and the activity coefficient of species in the condensed organic-phase, respectively. $P^o_L$ is the liquid vapor pressure and was estimated using a semi-empirical expression of the Clausius-Clapeyron equation (Eq. 2):

$$\ln(\frac{P_L^0}{760}) = -\frac{\Delta S_{vap}(T_b)}{R}[1.8(\frac{T_b}{T}-1)-0.8(\ln(\frac{T_b}{T}))]$$ (Eq.2)

where $T_b$ was the boiling temperature of different species, which was estimated using a previously described fragmentation method (Stein et al., 1994; Johnson et al., 2006). $\Delta S_{vap}$ (T$_b$) was the vaporization entropy change at T$_b$, which was estimated using the Trouton-Hildebrand-Everett rule with corrections for polar compounds and compounds with hydrogen-bonding capacity (Baum 1997). The concentration of species $j$ in the condensed organic-phase ($F_{j,om}$) can be calculated as the following equation (Eq. 3):

$$F_{j,om} = M_{om} \times (K_{p,j} \times A_j)$$ (Eq.3)

where $M_{om}$ is the total mass concentration of each condensed organic material from gas-particle partitioning, $A_j$ is its gas-phase concentration, and $K_{p,j}$ is its partitioning coefficient of species $j$ (Johnson et al., 2006).

The partitioning process was dynamically represented as an equilibrium between absorption and desorption, as described by Kamens et al. (1999). Briefly, the species-dependent $K_p$ values were defined in terms of absorption ($k_{in}$) and desorption ($k_{out}$) rate coefficients, with $K_p = k_{in}/k_{out}$. The value for $k_{in}$ ($k_{in}$ = 6.2 × 10$^{-3}$ m$^3 \cdot \mu$g$^{-1} \cdot$s$^{-1}$) was configured as suggested by Johnson et al. (2006). Thus, the $K_p$ could be expressed in terms of $k_{out}$. More detailed description of the equations and parameters are given by Johnson et al. (2005, 2006) and Kamens et al. (1999).

The above gas-particle partitioning of low volatility compounds formed by the gas-phase oxidation of VOCs and other precursors (Aumont, et al., 2012; Lee-Taylor et al., 2011) was configured in the model to estimate the SOA formation. However, the recent experimental results suggested that the formation of SOA in laboratory chambers may be suppressed due to losses of SOA to chamber walls, which leads to

underestimates of SOA in air-quality and climate models (Matsunaga and Ziemann
2010; Zhang et al., 2014). Therefore, to consider the wall loss of SOA, the average wall
loss rate coefficient of $6 \times 10^{-5}$ s$^{-1}$ was adopted in the model configuration according to
previous studies on the basis of calculated organic material using an assumed density
of 1g·cm$^{-3}$ (Johnson et al., 2004, 2005). In addition, the wall loss of other gaseous
compounds (O$_3$, NO$_2$ and HNO$_3$) were implemented in the box model with the average
parameters of $3 \times 10^{-6}$ s$^{-1}$, $1.15 \times 10^{-5}$ s$^{-1}$ and $8.2 \times 10^{-5}$ s$^{-1}$, respectively. The detailed
information for the calculation of above parameters was provided in Bloss et al. (2015).

*2.3. Partitioning and reactions of gas-phase dicabonyls on particles*
The partitioning and reactions of dicarbonyls in the aerosol aqueous phase may
involve both irreversible and reversible processes (Ervens and Volkamer, 2010). In the
present study, we follow the mechanism proposed by Knote et al. (2014) and consider
the reversible partitioning in aqueous phase, the irreversible volume reactions and
irreversible surface uptake processes in our model.
The reversible partitioning of Gly and Mgly on aerosols aqueous phase is usually
described by the Henry's law equilibrium (Kampf et al., 2013) (Eq.4):
$[Gly(Mgly)]_{liquid} = K_H \times [Gly(Mgly)]_{gas}$   (Eq.4)
However, hydration of carbonyls function groups and salt-Gly interactions could have
significant influences on the $K_H$ value of Gly (Kampf et al., 2013; Waxman et al., 2015),
and an effective Henry's law coefficient expressed by Eq.5 was often used.
$$K_{H,effective} = \frac{K_{H,water}}{10^{(-0.24 \min(12.0,(C_{as}+C_{an})))}}  \ (Eq.5)$$
where the $C_{as}$ and $C_{an}$ represent the concentrations of ammonium sulfate and nitrate.
The detailed information on each parameter in these equations have been provided in
Kampf et al. (2013), Waxman et al. (2015) and the supplementary of the present study.
As variations were found for the value of $K_{H,effective}$ under different concentrations of
ammonium sulfate and nitrate in previous studies (Knote et al., 2014; Kampf et al.,
2013; Erverns and Volkamer, 2010), the $C_{as}$ and $C_{an}$ were calculated every hour in the

present study from the measured ammonium sulfate (and ammonium nitrate) concentrations (mol m⁻³) divided by aerosol liquid water content (ALWC, kg m⁻³), which were determined by the aerosol inorganics model (AIM-IV, http://www.aim.env.uea.ac.uk/aim/model4/model4a.php) with inputs of the observed parameters (*e.g.*, ambient relative humidity, temperature, and the moles of each ion) at the Heshan site (Chang et al., 2019).

The reversible formation of monomer (*i.e.*, glyoxal, glyoxal monohydrate, and glyoxal dihydrate) and oligomers are considered with the two important reservoirs (*i.e.*, monomer and oligomer pools, represented as pool1 and pool2) (Knote et al. 2014). The variations of the glyoxal monomer ($[Gly_{p1}]$) and oligomer concentrations ($[Gly_{p2}]$) with time can be represented by the following equations (Erverns and Volkamer, 2010; Kampf et al., 2013; Knote et al., 2014):

$$\frac{d([Gly_{p1}])}{dt} = \frac{1}{\tau_1} \times (Gly_{p1,eq} - Gly_{p1}) \quad (Eq.6)$$

$$\frac{d([Gly_{p2}])}{dt} = \frac{1}{\tau_2} \times (Gly_{p2,eq} - Gly_{p2}) \quad (Eq.7)$$

$$\frac{Gly_{p2,eq}}{Gly_{p1,eq}} = K_{olig} \quad (Eq.8)$$

The equilibrium partitioning between monomers and oligomers was presented as $K_{olig}$ (Eq.8). The definition and configuration of each parameters above were provided in the supplementary (Section S2) according to Knote et al. (2014) and Kampf et al. (2013).

In addition, three irreversible pathways of Gly, including 1) the ammonium-catalyzed volume pathway, 2) the OH-reaction volume pathway, and 3) the irreversible surface uptake, were parameterized in the model (Knote et al., 2014; Ervens and Volkman 2010). The ammonium-catalyzed reactions, with rate constant depending on both particle acidity (pH) and the activity of the ammonium ion ($a_{NH4+}$), were parameterized as follows when the monomer and oligomer concentrations were in equilibrium (Eq. 9):

$$K = 2 \times 10^{-10} \times \exp(1.5 \times a_{NH_4}) \times \exp(2.5 \times pH) \times Gly_{p1} \quad (Eq.9)$$

This parameterization was configured based on the assumption that only total
concentration in the monomer pool was the only particulate glyoxal available to the
ammonium-catalyzed reaction as the reversibly formed oligomers do not evaporate
easily (Knote et al., 2014; De Haan et al., 2009; Noziere et al., 2008).
For OH pathway, the gas-phase OH was in equilibrium with liquid-phase OH by
a Henry's law constant ($K_{I,OH}$= 25 M atm$^{-1}$) with the consideration of the "salting-in"
impact (Ervens and Volkamer 2010), and constant of reactions between OH and Gly
was $1.1 \times 10^{-9}$ M$^{-1}$ s$^{-1}$ (Buxton et al., 1997). As suggested by Knote et al. (2014), the Gly
concentration available to the OH-reaction pathway was the total glyoxal concentration
in the monomer pool.
Surface-controlled irreversible uptake of Gly has been widely employed in
different modeling studies (Ervens et al., 2011; Li et al., 2014; Liu et al., 2007), was
parameterized as follows (Eq.10):
$$K_r = -\frac{\gamma_{gly(mgly)} \times S_{aw} \times v_{gly(mgly)} \times C^*_{gly(mgly)}}{4} \quad \text{(Eq.10)}$$

where $C^*$ and $v$ are the gas-phase concentration and mean molecular velocity,
respectively. $\gamma$ represents the uptake coefficient for Gly and Mgly. Here we use the
surface uptake coefficients ($\gamma_{gly}$= $1.0 \times 10^{-3}$ and $\gamma_{Mgly}$= $2.6 \times 10^{-4}$) to account for the
irreversible surface uptake of Gly and Mgly, respectively. It is noted that the surface
uptake coefficient of Gly was configured according to the results of uptake kinetics
experiments from Schweitzer et al. (1998), which has been used in the model simulation
of Gly in the previous PRD study (Li et al., 2014). On the other hand, the surface uptake
coefficient of Mgly was obtained via scaling to glyoxal uptake coefficient by the
relative Henry's low coefficient suggested by Pye et al. (2017). $S_{aw}$
($S_{aw}$=$S_a \times f$(RH)=$S_a \times (1+a \times$(RH)$^b$)) is the RH corrected aerosol surface area density (Li
et al., 2014). The value for $a$ (2.06) was configured as those suggested previously (Liu
et al., 2007), while the dry aerosol surface concentration ($S_a$) was obtained from the
measurement at the Heshan site (Yun et al., 2018). In this study, the mean molecular
velocities of Gly were calculated by the HyperPhysics model (http://hyperphysics.phy-

astr.gsu.edu/hbase, last access date: 06 June 2019). The carbonaceous and insoluble components were considered as an aqueous shell for aerosols, whereas the aerosol surface was fully covered with an aqueous layer (Li et al., 2015).

On the other hand, though heterogeneous processes of Mgly are similar to those of Gly, some difference between these two species were found. The Henry's law constant for Mgly is not as effective as that for Gly. Hence, a Henry's law constant ($3.7 \times 10^3$ M atm$^{-1}$) for Mgly we used (Zhou and Mopper 1990). In fact, Kroll et al. (2005) suggested that no obviously aerosol growth was observed from gas-phase Mgly presumably because of its more stable (less electron deficient) ketone moiety, and a recent study indicated that less Mgly would partition into the aerosols than expected according to Henry's law (Waxman et al., 2015). In addition, the surface uptake coefficient ($\gamma_{Mgly} = 2.6 \times 10^{-4}$) suggested by Pye et al. (2017) is lower than that extracted from the chamber study (De Haan et al., 2018), which reported the value of $\gamma_{Mgly}$ could increase to $3.7 \times 10^{-3}$ at 95% RH and even larger than Gly in a high relative-humidity environment ($\geq$95%). However, they also figured out that treating the surface uptake of Mgly on aerosols as an irreversible pathway could probably overestimate its positive effect for SOA formation via heterogeneous processes, because ~20% of SOA which were formed from Mgly via aqueous processes would further hydrolyze.

### 2.4. Model scenarios

According to the discussion above, it could be seen that the heterogeneous processes we described for Gly was more complicated than that for Mgly, as the parameterization for the sink of Gly from laboratory and model studies were more robust. Therefore, the present study put more emphasis on the evolution of Gly for better understanding and evaluating the effects of the different sink pathways on dicarbonyls and its influence on SOA formation. Table 1 provides detailed information regarding all the model scenarios for the simulation of Gly, while the model scenarios for Mgly are also given in Table S1 in the supplementary.

Table 1. Model scenarios used for gas-phase Gly

| Scenarios | Description | Purpose |
|---|---|---|
| INITIAL | Default MCMv3.2, without considering the reversible and irreversible uptake of Gly and the gas-particle partitioning of other oxidation products | Base run |
| scenario 1 | As INITIAL, also considers ammonium-catalyzed reactions of Gly through monomers pool 1 without the reversible formation of oligomers pool 2. | Investigating the influence of Ammonium reactions on the destruction of Gly |
| scenario 2 | As scenario 1, also considers OH reactions of Gly through monomers pool 1 without the reversible formation of oligomers pool 2. | Investigating the influence of OH reactions for the destruction of Gly |
| scenario 3 | As scenario 2, and considers the aqueous oligomers formation (pool 2) and revisable process with monomers (pool 1). | Investigating the "salting in" impact |
| scenario 4 | As scenario 3, and considers surface uptake by aerosols of Gly with the uptake coefficient of $1\times10^{-3}$ suggested by Li et al. (2014). | Investigating the influence of surface uptake |

360

In this study, hourly observation data of CO, $SO_2$, NO, $NO_2$, $O_3$, NMHC and meteorological parameters were used as input and constraints in the model. By taking the NMHC species incorporating in the MCM mechanism into account (MCM website, http://mcm.leeds.ac.uk/MCM/roots.htt, access date: 22 June 2020), observations of total 44 NMHC species, including 18 alkanes, 11 alkenes, ethyne and 14 aromatics were used as input for the model simulation (Table S2 in the supplementary). The selected NMHCs contributed about 98% and 99% to the total mixing ratios and photochemical reactivities of all measured NMHCs at the Heshan site. Furthermore, the selected VOCs are the major precursors for Gly, Mgly, photochemical $O_3$ and SOA (Ding et al., 2016, 2017; Li et al., 2014; Lou et al., 2010; Yuan et al., 2013), and have been frequently used to drive box model for studies on SOA, photochemical $O_3$ and photochemical reactivity (Hofzumahaus et al., 2009; Lee-Taylor, et al., 2011).

The photolysis rates, which were not measured, were modified in the model using the photon fluxes from the Tropospheric Ultraviolet and Visible Radiation (TUV-v5) model (Madronich and Flocke 1997) according to the sampling location and modeling period. Model simulation on Gly and Mgly was performed on January 07-08, 2017, when both daily Gly and Mgly data were available, with 00:00 LT (local time) as the

initial time. Before the simulation, the model was pre-run for 5 days using the observed
variability of the input species during the whole sampling period to achieve a steady
state for the unmeasured species with a short lifetime, *i.e.,* OH and $HO_2$ radicals (Xue
et al., 2014a, b).

382        In this study, the simulation on the diurnal variations of OH and $HO_2$ was

performed well, with peak values at noon, consistent with those measured and
simulated in PRD (Hofzumahaus et al., 2009 and related papers; Tan et al., 2019). The
simulated mean mixing ratios of OH and $HO_2$ radicals from the model in the present
study were ~$1.6 \times 10^6$ molecule·$cm^{-3}$ and ~$3 \times 10^7$ molecule·$cm^{-3}$, which are comparable
to the winter observations at Beijing, Tokyo, and New York (Kanaya et al., 2007; Ren
et al., 2006; Ma et al., 2019), and lower than the measurement and simulation values in
summer (e.g., July) or autumn (e.g., October to November) in the PRD region (Table
S3 in the supplementary) (Hofzumahaus et al., 2009; Tan et al., 2019). Note that the
variations of simulation results in the present study and those observation results in
previous studies in PRD may be associated with differences in the levels of $O_3$ and its
precursors, different photolysis rates, and to a lesser extent, meteorological conditions
(Hofzumahaus et al., 2009). The higher OH and $HO_2$ mixing ratios were expected in
summer and autumn than winter due to the stronger solar radiation and higher
temperature, as well as the variations of $O_3$ and its precursors in different sites, though
the measurement of OH/$HO_2$ radicals has been very challenging, and significant
uncertainties still exist in the measurement values of the radicals (Hofzumahaus et al.,
2009; Tan et al., 2019). Furthermore, the comparison between the simulation of a box
model and observation results suggested that the higher observed mixing ratios of OH
and $HO_2$ radicals were related to an unidentified source of OH at the backgarden site of
PRD in summer of 2006, while the comparison between the observed OH/$HO_2$
variations and those calculated from the parameterization of $HO_x$ ($HO_x$ = OH + $HO_2$)
production and destruction indicated a missing OH source of 4-6 ppbv·$h^{-1}$ and an
unknown $RO_2$ loss at the Heshan site in autumn of 2014.

406        In addition to the simulation of OH and $HO_2$ radicals, as there were no direct

measured SOA data in this study (Chang et al., 2019), the model performance was

evaluated by the comparison between the model simulated SOA with those calculated using the EC (elemental carbon)-tracer method, and by the comparison between the simulated and observed concentrations of other secondary products, which have been provided in detail in the supplementary (Section S3). For example, the simulated concentration of SOA was about 85% of those calculated by the EC-tracer method based on the observed hourly data (Chang et al., 2019). Furthermore, the simulated concentrations of acetic acid, formic acid and pyruvic acid were close to those observed at the Heshan site, accounting for ~80%, 70% and 88% of observed values for acetic acid, formic acid and pyruvic acid, respectively. The results confirmed that secondary formation was the dominant source of above species at the Heshan site, and suggested that the PBM-MCM model could provide robust performance on simulating the abundance of above secondary species and SOA.

### 2.5. Model uncertainty

Uncertainties in the simulation of Gly and Mgly by the model were noted. The total model errors could be calculated conservatively from 1) the uncertainties in the measurement of trace gases and NMHCs; 2) the measured data of meteorological parameters, *i.e.,* temperature $T$, pressure $P$, and the calculated photolysis frequencies $J$ based on meteorological conditions; 3) reaction rate constants $k$; and 4) the dry deposition. In this study, following Li et al. (2014) and Lu et al. (2013), the uncertainty factors for the above parameters were adopted as suggested previously (Table S4 in the supplementary), and all parameters were divided into three groups (*i.e.,* physical parameters, radical and trace gas concentrations, and reaction rate constants of non-photolytic reactions). Each parameter was multiplied by its uncertainty factor first, and the gaussian error propagation was then applied within each group. We run the model $n$ times ($n$ is the number of parameters considered). The mean diurnal variation of the uncertainty of modeled Gly and Mgly is shown in Figure S1. The total uncertainties of the modelled Gly and Mgly were both estimated to be around 39% with the contributions from radical and trace gas concentrations (~19%), physical parameters (~13%) (included photolysis frequencies, deposition lifetime, $T$, etc.) and reaction rate constants of non-photolytic reactions (~7%), respectively.

### *3. Results and Discussion*

### *3.1. Comparison between the simulation and observation*

In this study, the simulated Gly and Mgly were secondarily formed from the oxidation of their VOC precursors. Therefore, before the comparison between the simulation and observation results, the contributions of primary and secondary sources to the measured Gly and Mgly were preliminarily estimated by a correlation-based source apportionment method suggested by previous studies (Friedfeld et al., 2002; Yuan et al., 2013). Table S5 in the supplementary shows linear regression coefficients and relative source contributions of Gly and Mgly. It was found that the contributions from primary sources (3.46% and 3.51% for Gly and Mgly, respectively) were significantly lower than those from secondary sources (96.14% and 96.44%, respectively), confirming that observed Gly and Mgly in the present study were mostly related to secondary formation.

The simulated Gly and Mgly from the photochemical box model under different scenarios were examined and compared with the observation. The predicted Gly and Mgly from in-situ formation in the INITIAL scenario was found to generally overpredict the mixing ratios of Gly and Mgly, and were about 3.3 and 3.5 times of the observed concentrations at the Heshan site, respectively. Similar overestimation has been reported in previous modeling studies, for example, the simulations with only the MCM gas-phases schemes overpredicted the Gly concentration by factors of 2-6 in both urban Mexico City (Volkamer et al., 2007) and a semi-rural site of the PRD region (Li et al., 2014). The significant overestimation in simulation results indicated that there were important loss pathways for Gly and Mgly other than the oxidation reactions (*e.g.,* by OH and $NO_3$ radicals). A sensitivity analysis was firstly conducted with twice the dilution rate (and deposition velocities) in INITIAL scenario, which resulted in the reduction of modeled concentrations of Gly and Mgly by 9.2% (3.2%) and 7.9% (2.8%), respectively (Table S6 in the supplementary). Though these enlarged rates were higher than the upper limits of the dilution rate and deposition velocities in previous studies (Fan et al., 2011; Wang et al., 2013; Li et al., 2014), the predicted mixing ratios of Gly and Mgly were still three times higher than the measured levels, suggesting that the

dilution and dry deposition configuration could not be the main causes for the
overestimation of modeled Gly and Mgly mixing ratios (Volkamer et al., 2006, 2007).

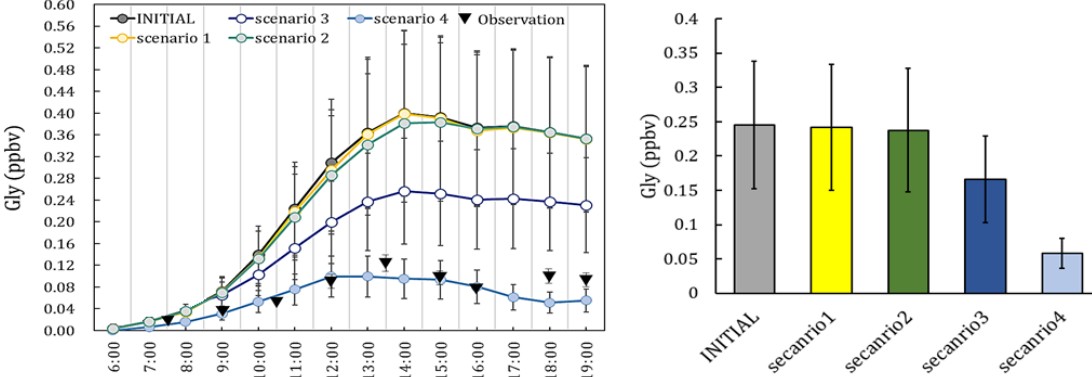

Figure 2. The observation data, the concentrations and the daily average concentration of Gly
predicted from the different scenarios.
To identify the causes of discrepancy and improve model performance, sensitivity
analysis with different heterogenous mechanisms incorporated in the model scheme (as
scenarios listed in Table 1) was conducted. The average diurnal patterns of Gly
simulated by different model scenarios are showed in Figure 2. It can be seen that in
the early morning (*i.e.*, 0600 to 0800 LT, local time), the predicted mixing ratios of Gly
in different scenarios were comparable to the observation. In contrast, the difference
became larger from 0900 LT onwards, though the predicted peaks of Gly by different
model scenarios were all presented at the early afternoon, following by a slow decrease
in the late afternoon. The model results from Scenario 4 was more consistent with the
observational data. The relative changes of modeled Gly concentrations by adding
additional heterogeneous processes to the model scenario INITIAL (*i.e.*, model
scenarios 1-4) were summarized in Table S7 in the supplementary.
On average, by including additional irreversible and reversible pathways, the
modeled Gly concentrations during daytime (06:00-19:00) decrease by 72.3% of the
values predicted by the INITIAL scenario, and a significant decrease of Gly
concentration occurred by adding the effect of surface uptake pathway (*i.e.*, model
scenarios 3-4). Similar results have been obtained in a previous study in summer in the
PRD region (Li et al., 2014), which found that the simulated Gly concentration decrease
significantly (~45 %) in the daytime (*i.e.*, 0600 to 1900 LT) when heterogeneous uptake
process was considered with the incorporation of a single uptake coefficient.
The contribution of different heterogeneous sink pathways is calculated based on

the scenario 4 (Figure S2). The irreversible pathways of Gly (*i.e.*, surface uptake by aerosols, OH and ammonium reactions) accounted for 67.3% of the total sink of Gly, among which the surface uptake was a dominant pathway (62%) comparing to the ammonium and OH reactions (2.4% and 2.9%, respectively). The reversible pathway made a relatively lower contribution to the total sink of Gly (32.7%).

In addition, the heterogeneous irreversibly and reversibly pathways of Mgly was also investigated in the sensitivity analysis (*i.e.*, model scenarios M1-2, listed in Table S8 in the supplementary), and the predicted Mgly concentrations during daytime (0600-1900 LT) decreased by 73.0% of the values estimated in INITIAL model scenario. The surface uptake pathway in the scenario M2 was found to be the most important heterogeneous pathway for the loss of Mgly, and contributed to 64.1% of the total heterogeneous sink of Mgly. The contributions of the reversible pathway to the destruction of Mgly was around 35.9% (Figure S3).

Overall, by incorporating a more detailed heterogeneous processes of Gly and Mgly, the results of scenarios 4 and M2 provided better agreement between the modeled and measured Gly and Mgly. The results demonstrated the significance of heterogeneous uptake processes on the destruction of Gly and Mgly, and adopting the irreversible/reversible pathways (*i.e.*, the reversible partitioning, volume reactions, and the surface uptake) could reasonably reproduce the variations of Gly and Mgly at the Heshan site in the PRD region.

### *3.2. Process analysis on the production and destruction of Gly and Mgly*

The scenarios 4 and M2 simulation with the best agreement with measurement were further analyzed to investigate the photochemical budget of Gly and Mgly at the Heshan site, respectively (Table 2 and Table S1). It was found that OH oxidation of aromatics was the most important contributor for the Gly and Mgly production, with mean contributions of ~80% and ~94%, respectively. Among all the aromatic precursors, toluene and *m,p*-xylene were the two major precursors for the formation of Gly and Mgly, with total contributions of ~43% and ~56% of Gly and Mgly formation, respectively. In contrast, because of the relatively low photochemical reactivity, benzene and alkanes had lower contributions to the formation of Gly and Mgly, although they can travel a long distance and contribute to secondary Gly and Mgly in

areas far from their emissions (Lv et al., 2019). Different from previous studies that
found the isoprene as the key precursor for Gly and Mgly formation (Li et al., 2014;
Lou et al., 2010), the contributions of isoprene oxidation at the Heshan site in the
present study were much lower than that of aromatics, with only mean contributions of
~2% and ~3%, respectively. It can be attributed to the lower mixing ratios of isoprene
(*i.e.*, 70 ± 10 pptv) observed at the Heshan site because of the lower temperature in
winter, comparing to the much higher concentration observed during summer (average
of ~1 ppbv and maximum of ~ 4 ppbv in the afternoon) in the rural and forest areas in
this region (Li et al., 2014; Lou et al., 2010).

538        In addition, the relative contributions of different loss pathways of Gly and Mgly,

including physical processes (vertical dilution and dry deposition), reaction with
radicals (e.g., OH and $NO_3$), and the heterogeneous processes of Gly and Mgly on
aerosols at daytime were also estimated from the PBM-MCM model results (Table 2).
Consistent with previous studies (Atkinson and Arey, 2003; Ervens et al., 2011),
heterogeneous processes were the most important pathway for the destruction of Gly
and Mgly (with contributions of ~62% during daytime), followed by photolysis (with
contributions of ~26% and ~25%, respectively). It should be noted that the oxidation
of Gly and Mgly by $O_3$ was not considered in this study as the reaction rate constants
of Gly and Mgly with $O_3$ are < 3 and < 6 × $10^{-21}$ $cm^3 \cdot molecule^{-1} \cdot s^{-1}$, respectively,
which are 6 order of magnitude lower than the reaction rate constants with $NO_3$
(which >1 and > 2 × $10^{-15}$ $cm^3 \cdot molecule^{-1} \cdot s^{-1}$, respectively), and are 9 order of
magnitude lower than the reaction rate constants with OH (9 and 13 × $10^{-12}$
$cm^3 \cdot molecule^{-1} \cdot s^{-1}$ for the reactions of Gly and Mgly with $O_3$, respectively) (Mellouki
et al., 2015). Therefore, we believe that the influence of $O_3$ on the removal of Gly and
Mgly was negligible (Mellouki et al., 2015). Furthermore, there were few
parameterizations for the reaction mechanism of Gly/Mgly with $O_3$ due to their low
reaction rates with $O_3$.

556        On the other hand, at nighttime, only the heterogenous processes made the main

contribution to Gly and Mgly destruction, with contributions higher than 90% to the
total destruction of Gly and Mgly at night (Table S9 in the supplementary), consistent
with previous studies (Washenfelder et al., 2011; Gomez et al., 2015). The lower
contributions of Gly and Mgly with radicals were mainly because of the low OH
concentration at night and their relatively lower reactivities with $NO_3$ radical (e.g., the
reaction rate constants of Gly/Mgly with $NO_3$ are ~1000 times lower than those with
OH radical) (Calvert et al., 2011; Mellouki et al., 2015).

Table 2 Production and destruction of Gly and Mgly from model simulation at daytime

| Precursor | Oxidant | Gly | | Mgly | |
|---|---|---|---|---|---|
| | | Molar yield (%)[a] | Contribution (%) | Molar yield (%)[a] | Contribution(%) |
| **Aromatics** | | | | | |
| benzene, % | OH | 32 | 4.94 | - | - |
| toluene, % | OH | 30.6 | 23.41 | 21.5 | 23.80 |
| *m, p*-xylene, % | OH | 25.2 | 19.22 | 35.1 | 32.08 |
| *o*-xylene, % | OH | 12.7 | 15.04 | 33.1 | 14.49 |
| 1,2,4-trimethylbenzene, % | OH | 7.2 | 1.40 | 27.2 | 5.98 |
| 1,2,3-trimethylbenzene, % | OH | 7.8 | 1.43 | 15.1 | 4.54 |
| 1,3,5-trimethylbenzene, % | OH | -[c] | - | 58.1 | 13.21 |
| ethylbenzene, % | OH | 55 | 6.62 | - | - |
| *p*-ethyltoluene, % | OH | 31.9 | 5.45 | - | - |
| *m*-ethyltoluene, % | OH | 7.9 | 1.52 | - | - |
| *o*-ethyltoluene, % | OH | 8 | 0.51 | - | - |
| **Sum** | | | 79.54 | | 94.10 |
| Alkanes | | | | | |
| propane, % | OH | - | - | 11 | 0.73 |
| > C3 alkanes[b],% | OH | 1 | 0.19 | 3.2 | 0.71 |
| Sum | | | 0.19 | | 1.44 |
| **Alkenes** | | | | | |
| isoprene,% | OH | 6.2 | 0.43 | 25 | 0.57 |
| | $NO_3$ | 43.7 | 1.34 | 37.8 | 2.83 |
| | $O_3$ | 4 | 0.20 | - | - |
| ethene,% | OH | 5.7 | 1.08 | - | - |
| | $O_3$ | 0.44 | 1.15 | - | - |

| | | | | | |
|---|---|---|---|---|---|
| > C2 alkenes [b],% | OH | - | - | 7.7 | 1.06 |
| propene, % | $O_3$ | 8.3 | 1.01 | - | - |
| 1-pentene, % | $O_3$ | 2 | 0.73 | - | - |
| Sum | | | 5.94 | | 4.46 |
| Acetylene | OH | 63.5 | 14.33 | - | - |
| **Loss pathyways** | | | | | |
| photolysis, % | | | 26.2 | | 25.1 |
| $NO_3$,OH-reaction, % | | | 4.06 | | 7.87 |
| dry deposition, % | | | 2.23 | | 1.73 |
| dilution, % | | | 5.71 | | 3.30 |
| heterogeneous [d], % | | | | | |
| Irreversible processes, % | | | 41.0 | | 39.8 |
| Reversible processes, % | | | 20.8 | | 22.2 |

[a] Molar yields were taken from previous studies (Fu et al., 2008) (Fick et al., 2003) (Nishino et al., 2010) (Calvert 2000; Volkamer et al., 2006).

[b] >2 alkenes (include 3 alkenes) and >3 alkanes (include 17 alkanes) are represented in this study as a single lumped species (Lv et al., 2019).

[c] "–" not applicable.

[d] Considered both irreversible and reversible parameterizations of the aerosol sinks (*i.e.*, scenario 4 and M2 in the supplementary).

### *3.3 Implications for secondary organic aerosol formation*

By incorporating both the traditional gas-particle partitioning (of VOC oxidation products) and the heterogeneous processes (of Gly and Mgly) into the model, we investigated the contributions of different mechanism in SOA formation through sensitivity analysis. The contributions of VOC oxidations to SOA formed from gas-particle partitioning ($SOA_{gp}$) and SOA formed from heterogeneous processes of Gly and Mgly ($SOA_{het}$) were quantified.

On the other hand, only based on the $SOA_{gp}$ formation scheme, the relative importance of each VOC precursor in $SOA_{gp}$ formation was further evaluated to provide a complete picture for $SOA_{gp}$ formation and its relationship with precursors. As with $O_3$ formation, the roles of individual VOC precursors in $SOA_{gp}$ formation were calculated using relative increment reactivity ($RIR_{SOAgp}$) method, which have been widely used to present the percentage change in the production of secondary products per percent change in precursors. The $RIR_{SOAgp}$ of a specific precursor $X$ at site Z is given by Eq. 11:

$$RIR_{SOA_{gp}}^{Z}(X)=\frac{[P_{SOAgp}^{Z}(X)-P_{SOAgp}^{Z}(X-\Delta X)]/P_{SOAgp}^{Z}(X)}{\Delta Z(X)/Z(X)} \quad \text{(Eq.11)}$$
where $Z(X)$ represents the measured concentration of precursor $X$, including the
amounts emitted at the site and those transported to the site, and $\Delta X$ is the change in the
concentration of precursor $X$ caused by a hypothetical change $\Delta Z(X)$ (10% $Z(X)$ in this
study). Here, $P_{SOA_{gp}}^{Z}(X)$ represents the SOA$_{gp}$ formation potential. A large positive
$RIR_{SOAgp}$ value of a specific precursor suggests that SOA$_{gp}$ formation could be
significantly decreased if the emissions of this precursor were controlled. Figure 3
depicts the top 10 VOC precursors with high $RIR_{SOAgp}$ values at day time. Both $m,p$-
xylene and $o$-xylene had the highest $RIR_{SOAgp}$ value (~0.35), followed by toluene (~0.2)
and ethylbenzene (~0.06). As $m,p$-xylene, $o$-xylene and toluene can also have a
significant impact on dicabonyls production, they are likely to make a noticeable
contribution to both SOA$_{gp}$ and SOA$_{het}$ formation.

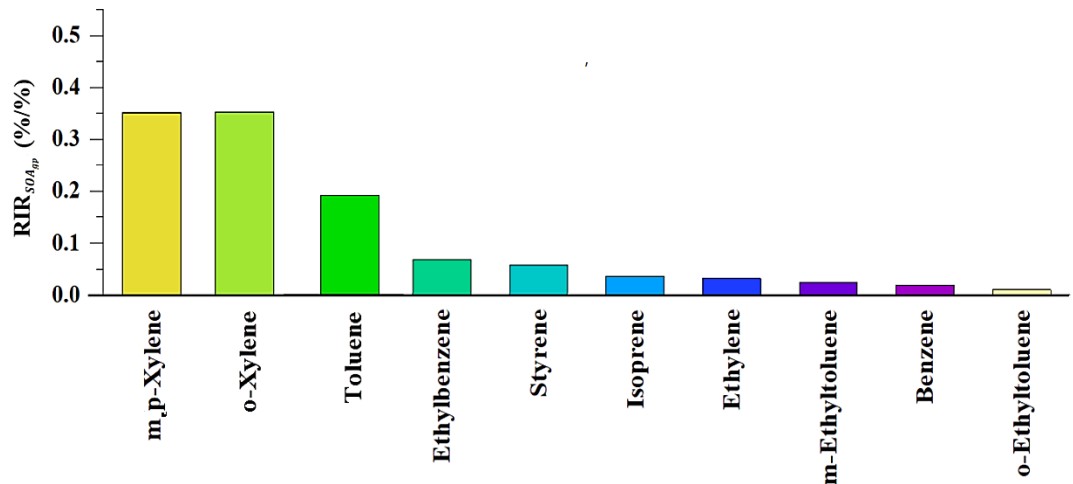


Figure 3. Top 10 VOC precursors with high $RIR_{SOAgp}$ values at daytime.
The SOA production from Gly and Mgly was further explored by the model
simulation with and without the consideration of Gly and Mgly. It was found that by
incorporating the evolution of Gly and Mgly, the SOA production has been improved
apparently from 1.83 to 2.47 $\mu g/m^3$. The total contribution of the Gly and Mgly
contributed ~26% to the simulated SOA concentrations, of which ~21% was from the
heterogeneous processes of Gly and Mgly (SOA$_{het}$), further demonstrating that the
heterogeneous processes have significant influences on the SOA formation from Gly
and Mgly.
To further highlight the roles of heterogeneous processes of Gly and Mgly on the
SOA production (SOA$_{het}$) and to evaluate the contributions of different VOCs, the

average diurnal variations of $SOA_{het}$ concentration formed from the heterogeneous processes of Gly and Mgly were showed in Figure 4. Both $SOA_{het}$ (Gly) and $SOA_{het}$ (Mgly) concentrations presented photochemistry-driven diurnal patterns, and started to increase in the morning before reaching the maximum value (0.52 and 0.42 $\mu g/m^3$) at 1400 and 1200 LT, respectively. It is consistent with the diurnal pattern of $SOA_{gp}$, which could be formed from the oxidation of VOCs (including NMHCs and the gaseous oxidation of Gly and Mgly which were formed from the oxidation of NMHCs), due to the high photochemical reactivity at noon, which further converted to $SOA_{gp}$ through gas-particle partitioning. In general, $SOA_{gp}$ made a higher contribution to total SOA (78.6%) than $SOA_{het}$ (21.4%). Previous studies have indicated that the more abundant anthropogenic precursors than biogenic ones under $NO_x$ saturated environment could lead to greater contribution of $SOA_{gp}$ to total SOA despite that the oxidation of anthropogenic species (*i.e.*, aromatics) could lead to relatively higher yields of Gly and Mgly (Knote et al., 2014; Ervens et al., 2011). Ervens et al. (2011) has found that in areas with high concentrations of biogenic precursors at high relative humidity, the $SOA_{het}$ and $SOA_{gp}$ were equally important for total SOA, while in the anthropogenic dominated areas, the contribution of $SOA_{het}$ to the total predicted SOA mass was around 30%. Similarly, the formation of $SOA_{gp}$ and $SOA_{het}$ were both dominated by xylenes and toluene, contributing to ~74%, ~62% and ~69% of $SOA_{gp}$, $SOA_{het}$ (Gly) and $SOA_{het}$ (Mgly), respectively. Furthermore, *o*-xylene was the most important precursor to the $SOA_{gp}$ (~29%), but only contributed ~16% and ~13% to $SOA_{het}$ (Gly) and $SOA_{het}$ (Mgly) formation, respectively. The toluene and *m,p*-xylene made the most significant contributions to the $SOA_{het}$ (Gly) (~26%) and $SOA_{het}$ (Mgly) (~32%) formation, respectively, consistent with the chamber results from the oxidations of different precursors (Ervens et al., 2011).

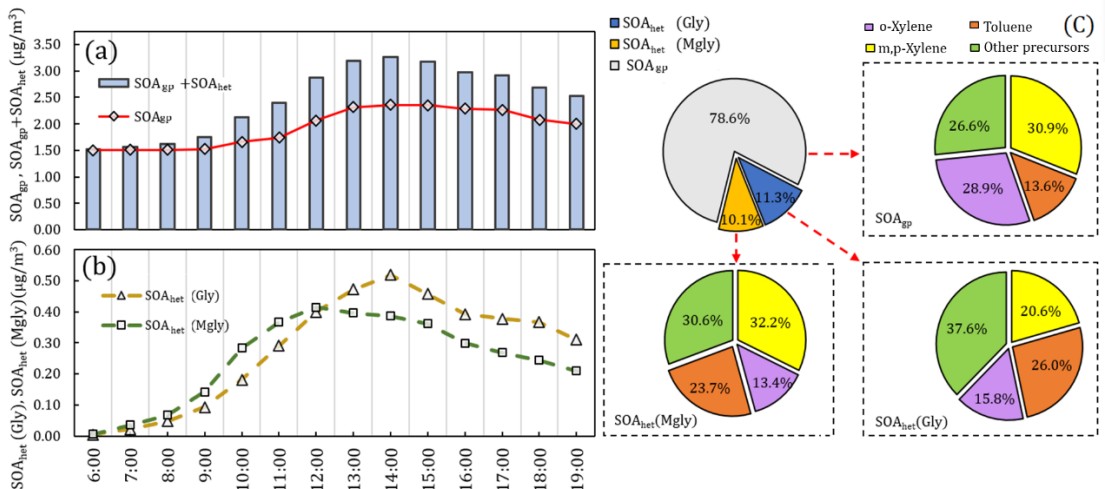


Figure 4. The average diurnal variations of $SOA_{gp}$, $SOA_{het}$ (Gly) (*i.e.*, $SOA_{het}$ formed from
heterogeneous processes of Gly), $SOA_{het}$ (Mgly) (*i.e.*, $SOA_{het}$ formed heterogeneous processes
from Mgly) and total SOA formation ($SOA_{gp}$ + $SOA_{het}$) were showed in Figure 4a and 4b. The
proportion of $SOA_{gp}$, $SOA_{het}$ (Gly) and $SOA_{het}$ (Mgly) in total SOA as well as the contribution
of VOCs precursors to $SOA_{gp}$, $SOA_{het}$ (Gly) and $SOA_{het}$ (Mgly) formation, were represented
in the pie charts in Figure 4c.

### 3.4 Comparison with previous studies in PRD
Previous studies have been conducted to investigate the evolution of Gly and/or
Mgly using observation and model simulation in the PRD region. However, one must
bear in mind that the mechanisms of the formation and evolution of Gly and/or Mgly
were more detailed in the present study, as previous studies in PRD only incorporated
the irreversible surface uptake process with a single coefficient for the heterogeneous
pathway for dicarbonyls. This may not reflect the current knowledge for the formation
and evolution of Gly and/or Mgly and their influence on SOA formation (Knote et al.,
2014; Waxman et al., 2015; Sumner et al., 2014).
Li et al. (2013a) used the regional air quality model CMAQ to investigate the
contributions of the aerosol surface uptake of Gly and Mgly to SOA formation in the
PRD region, and an uptake coefficient of $2.9 \times 10^{-3}$ was used for both Gly and Mgly in
the model. Interestingly, the results from their model were about 30% higher than that
in our study (Table S10 in the supplementary). For example, Li et al. (2013a) concluded
that SOA formed from the heterogeneous processes of dicabonyls may contribute
higher than 50% to the total SOA mass in the PRD region, while our study showed that
the contribution of $SOA_{het}$ to total SOA mass was ~21% (*i.e.*, ~11% of $SOA_{het}$ formed
from Gly; ~10% of $SOA_{het}$ formed from Mgly). In addition, the averaged concentration
of $SOA_{het}$ from Gly (0.28 µg/m$^3$) and Mgly (0.25 µg/m$^3$) in our study is one order of

magnitude lower than that in Li et al. (2013a) (*i.e.*, 2.33 and 2.51 μg/m$^3$, respectively). The discrepancy was mainly due to the different parameterizations of heterogeneous processes of dicarbonyls. The parameterization in the Li et al. (2013a) and other previous studies did not consider the reversible processes of dicarbonyls, but used one constant surface uptake coefficient to represent all the heterogeneous processes, which could result in bias in SOA formation if there are available aerosol surfaces without considering the influence of aerosols composition and phase state. Moreover, most of the previous studies using higher surface uptake coefficients intended to narrow the discrepancy between observed and simulated SOA mass without direct comparison between observed and simulated concentrations of dicarbonyls (i.e., Li et al., 2013a; Waxman et al., 2013; Fu et al., 2008; Vokalmer et al., 2007). For example, Knote et al. (2014) conducted a total of seven simulations to investigate the SOA formation from Gly over California. Their results showed that the SOA concentration in SIMPLE scenario (characterized by a single uptake coefficient of $3.3 \times 10^{-3}$) was an order of magnitude higher than that in HYBRID scenario (characterized by an uptake coefficient of $1.0 \times 10^{-3}$ and also considered more comprehensive parameterization of heterogeneous processes). In fact, if we only consider the surface uptake by aerosols for dicarbonyls using the same uptake coefficient for dicarbonyls ($2.9 \times 10^{-3}$) as Li et al., (2013a), the contribution of SOA$_{het}$ to total SOA mass would increase to 72% (*i.e.*, 37% of SOA$_{het}$ formed from Gly; 35% of SOA$_{het}$ formed from Mgly) (Table S10 in the supplementary). However, this configuration may not reflect the real evolution of dicarbonyls, resulting in the underestimation on the dicarbonyls concentrations (i.e., the simulated concentration is at least one order of magnitude lower than the observation) (Figure S4) and overprediction of the contribution of SOA$_{het}$ to total SOA mass in this study (~51%).

Table 3 compares the surface uptake coefficient derived from laboratory experiments and those used in different model simulation. It could be found that there was a large variation range for the surface uptake coefficients of Gly, while the studies on Mgly were still limited. For example, the laboratory experiment reported the surface uptake coefficients of Gly in the range of (0.8-6.6) and ($\leq 1$ - 9) $\times 10^{-3}$ on aqueous inorganic aerosols and cloud droplet/ice crystals, respectively (Volkamer et al., 2007; Loggio et al., 2005), and the coefficients were found to be $> 2.3 \times 10^{-3}$ for particles with high acidity (pH values within the range of -0.44 to -1.3) (Loggio et al., 2005). On

the other hand, Schweitzer et al. (1998) reported that the uptake coefficient of > 0.001 was only observed for lower temperature conditions, and the experimentally measured coefficient ranged from $(1.2 \pm 0.06) \times 10^{-2}$ to $(2.5 \pm 0.01) \times 10^{-3}$ on acidic solution (*i.e.*, 60-93 wt% $H_2SO_4$) at 253-273 k (Gomez et al., 2015; Zhang et al., 2015). It is suggested that more accurate and comprehensive parameterization of heterogeneous processes of dicabonyls still needs deeper exploration for further model development. The parameterization used in this study were mostly adopted from previous results, though it may still have limitations and uncertainties, the results of simulation at this site show better agreement with the observation.

Table 3 Surface uptake coefficient of Gly from laboratory experiments and used in the model simulation in the present and previous studies

| Coefficient | References |
|---|---|
| $(0.8\text{-}7.3) \times 10^{-3}$, on aqueous inorganic aerosols | Volkamer et al., 2007; Loggio et al., 2005; |
| $(\leq 1\text{ - }9) \times 10^{-3}$, and on cloud droplet/ice crystals | Volkamer et al., 2007; Loggio et al., 2005; |
| $(1.2 \pm 0.06) \times 10^{-2}$ - $(2.5 \pm 0.01) \times 10^{-3}$ on acidic solutions (*i.e.*, 60-93 wt% $H_2SO_4$ at 253-273 k) | Gomez et al., 2015; Zhang et al., 2015 |
| $3.3 \times 10^{-3}$ | Knote et al., 2014 and references therein; Waxman et al., 2013; Waxman et al., 2013 |
| $2.9 \times 10^{-3}$ | Fu et al., 2008 |
| $1.0 \times 10^{-3}$ | Knote et al., 2014 and references therein; this study; Li et al., 2014 |

*4. Conclusion*

A photochemical box model coupled with MCM (v3.2) (PBM-MCM) and further improvements on the evolution of semi- and non-volatility oxidation products to a condensed particle-phase, was used to investigate the production and heterogeneous processes of Gly and Mgly, as well as the SOA-precursor relationship at a receptor site (*i.e.,* the Heshan site) for the first time in the PRD region. Compared to the measurements, the initial model configuration overestimated the Gly and Mgly concentrations by a factor of 3.3 and 3.5, respectively. This discrepancy occurred largely due to the absence of irreversible uptake and reversible partitioning. Model investigation regarding the production of Gly and Mgly revealed that the oxidation of

aromatics by OH radicals was the most important contributor to the formation of Gly and Mgly, with mean contributions of ~80% and ~94%, respectively, with toluene and *m,p*-xylene acting as the most important precursors for Gly and Mgly. For SOA formation, the heterogeneous processes of Gly and Mgly probably can explain ~21% of SOA mass in PRD. Toluene and *m,p*-xylene were the main precursors for $SOA_{het}$ formation, while *o*-xylene was the most important precursor of $SOA_{gp}$. Overall, this study evaluated the formation and heterogeneous processes of Gly and Mgly in a polluted subtropical environment and highlighted the important role of intermediate products that are produced from photochemical oxidation of VOCs in SOA formation. The results of this study are expected to provide a better understanding of the evolution of VOC precursors, intermediate products, and heterogeneous process of the dicarbonyls, and the developed model modules can provide a robust tool for investigating SOA formation in the PRD and other regions in China.

*Author contributions*
In this study, the model was developed by ZL and QX. The whole structure for the paper was designed by ZL, XW and ZW. QX, ZL and ZW wrote the manuscript. ZW provided the observed data. All the authors have made substantial contributions to the work reported in the manuscript. ZL and QX contribute equally to this article.

*Data availability*
The underlying research data and the newly developed MCM scheme of Gly and Mgly in this study are available to the community and can be accessed by request to Zhenhao Ling (lingzhh3@mail.sysu.edu.cn) of Sun Yat-sen University.

**Competing interests**

The authors declare that they have no conflict of interest.

*Acknowledgements*
The authors thank the Collaborative Innovation Center of Climate Change, Jiangsu Province, and also thank Barbara Ervens for her constructive comments for the manuscript.

*Funding Sources*
This research has been supported by the National Key Research and Development Program of China (grant nos. 2017YFC0210106 and 2016YFC0203305), the National Natural Science Foundation of China (grant nos. 91644215, 41775114, 91744204), and Research Grant Council of the Hong Kong Special Administrative Region, China (grant nos. 15265516, 25221215 and T24/504/17). This work was also partly supported by the

Pearl River Science and Technology Nova Program of Guangzhou (grant no. 201806010146), the Fundamental Research Funds for the Central Universities (grant no. 19lgzd06), the Special Fund Project for Science and Technology Innovation Strategy of Guangdong Province (Grant No.2019B121205004).

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
