# Peer review of "Formation and sink of glyoxal and methylglyoxal in a polluted subtropical"

_Atmospheric Chemistry and Physics, 2020_

## Referee Comment (RC1) · Anonymous Referee #1 · 11 May 2020

This study investigating the contribution of GLY and MGLY to SOA based on observation and box modeling. Overall this study is of interesting based on the methods and results. However, improvements are needed for its publication. 1. The study used VOCs observations to drive the model with MCM mechanism, but there is no list of the VOCs. Are the species enough to drive the mechanism? If major species are missing, the chemistry may be messed up. 2. The SOA simulation was not validated, thus it is not clear if the contribution of GLY and MGLY are in realistic range. For example, the study may underestimate or overestimate SOA from other species. Also, SOA for-

mation pathways are far from accurate, if corrections such as wall loss are considered in this study? It is important to make sure that what recent findings regarding SOA formation have been considered, as the total SOA prediction would be much different. 3. The basis for heterogeneous reactions are the comparison of gas phase of GLY and MGLY concentrations, which may cause uncertainties. Without validating the components formed by GLY and MGLY in SOA, it is misleading that GLY and MGLY are actually converted to SOA. 4. When considering the budget, it is import to considering transport. It is suggested to exclude the effects of transport in/out or note the readers regarding the uncertainties.
* * *

---

## Referee Comment (RC2) · Anonymous Referee #2 · 11 May 2020

The paper investigates the importance of glyoxal (GLY) and methylglyoxal (MGLY) on SOA formation in the PRD region. The importance of GLY and MGLY on the SOA formation has been previously studied but this study investigated several different methods of modeling GLY and MGLY SOA, which provides additional information on how to model this important process. I have a few suggestions for the authors when they revise their paper: 1. The gas-phase concentrations of OH/HO2 are not constrained in the box model simulations. Instead, they are calculated using the box model. However, GLY and MGLY can be removed in the gas phase reactions with OH. Thus, it is

necessary to evaluate the predicted OH/HO2 concentrations to make sure the competing between gas/particle partitioning, which forms SOA, and the gas phase decay processes that reduce the SOA formation is correctly captured. 2. The impact of O3 on GLY and MGLY is not discussed. Looking at Figure 2, it is obvious that GLY and MGLY must decrease at night. GLY and MGLY can also react with O3. This is likely an import process that reduces GLY and MGLY at night, in addition to SOA formation from gas-to-particle partitioning and aqueous reactions. Since GLY and MGLY data are collected throughout the entire 24 hour period, it might be interesting to see how well the box model predicts GLY and MGLY at night with different SOA modeling approaches. The nighttime behavior of GLY and MGLY and their roles in SOA formation is not as clear as the daytime and should not be ignored in this study. 3. The other issue that I think should be addressed is the primary emissions of GLY and MGLY, since not all are produced secondarily. It seems that no emissions of primary GLY and MGLY are included in the box model simulations. The authors might want to discuss how this omission can impact their conclusions. 4. Modeling SOA formation from GLY and MGLY using the Master Chemical Mechanism has been discussed in a more complete regional air quality model that considers the transport and emissions of GLY and MGLY (Li et al. 2015). Another paper that discusses the importance of the GLY and MGLY on SOA formation is Ying et al 2015, which shows that including SOA from uptake of GLY/MGLY leads to significant improvement in predicated SOA in these southeast United Stated. These references appear to be neglected by the authors.

Li, J., Cleveland, M., Ziemba, L. D., Griffin, R. J., Barsanti, K. C., Pankow, J. F., and Ying, Q.: Modeling regional secondary organic aerosol using the Master Chemical Mechanism, Atmospheric Environment, 102, 52-61, 2015.

Ying, Q., Li, J., and Kota, S. H.: Significant Contributions of Isoprene to Summertime Secondary Organic Aerosol in Eastern United States, Environmental Science and Technology, 49, 7834–7842, 2015.

---

## Author Comment (AC1) · 21 Jul 2020

Zhenhao Ling et al. lingzhh3@mail.sysu.edu.cn

Response to Reviewers

Reviewer #1 This study investigating the contribution of GLY and MGLY to SOA based on observation and box modeling. Overall this study is of interesting based on the

methods and results. However, improvements are needed for its publication. Reply: We appreciate the two anonymous reviewers for their constructive criticisms and valuable comments, which were of great help in improving the quality of the manuscript. We have revised the manuscript accordingly and our detailed responses are shown below. All the revision is highlighted in the revised manuscript.

1. The study used VOCs observations to drive the model with MCM mechanism, but there is no list of the VOCs. Are the species enough to drive the mechanism? If major species are missing, the chemistry may be messed up. Reply: The reviewer's comment is highly appreciated. In this study, mixing ratios of 53 C2-C10 non-methane hydrocarbons (NMHCs), including 27 alkanes, 11 alkenes, ethyne and 14 aromatics were measured continuously from 2-8 January, 2017, which detailed information and statistics has been provided in our previous studies (Chang et al., 2019; Yun et al., 2018). By taking the species incorporating in the MCM mechanism into account (MCM website, http://mcm.leeds.ac.uk/MCM/roots.htt, access date: 22 June 2020), observations of total 44 VOC species, including 18 alkanes, 11 alkenes, ethyne and 14 aromatics used as input for the model simulation (Table 1). It should be noted that the selected NMHCs contributed about 98% and 99% to the total mixing ratios and photochemical reactivities of all NMHCs measured at the Heshan site. Furthermore, the selected VOCs, which were frequently used to drive box model for SOA, photochemical O3 and photochemical reactivity (Hofzumahaus et al., 2009; Lee-Taylor, et al., 2011), were the major precursors for Gly, Mgly, photochemical O3 and SOA (Ding et al., 2016, 2017; Hofzumahaus et al., 2009; Li et al., 2014; Lou et al., 2010; Yuan et al., 2013). Therefore, we believe that the selected species were enough to drive the PBM-MCM model and simulate the formation of Gly, Mgly and SOA in this study. Table 1. The NMHC Compounds which driven the PBM-MCM model No. Compound No. Compound 1 Ethane 23 1-Butene 2 Propane 24 cis-2-Butene 3 i-Butane 25 1-Pentene 4 n-Butane 26 trans-2-Pentene 5 i-Pentane 27 Isoprene 6 n-Pentane 28 cis-2-Pentene 7 2,2-Dimethylbutane 29 1-Hexene 8 2,3-Dimetylbutane 30 1,3-Butadiene 9 2-Methylpentane 31 Benzene 10 3-Methylpentane 32 Toluene 11 n-Hexane 33 Ethylbenzene 12 2-Methylhexane 34

m/p-Xylene 13 Cyclohexane 35 o-Xylene 14 3-Methylhexane 36 Styrene 15 n-Heptane 37 i-Propylbenzene 16 n-Octane 38 n-Propylbenzene 17 n-Nonane 39 m-Ethyltoluene 18 n-Decane 40 p-Ethyltoluene 19 Ethene 41 1,3,5-Trimethylbenzene 20 Propene 42 o-Ethyltoluene 21 Ethyne 43 1,2,4-Trimethylbenzene 22 trans-2-Butene 44 1,2,3-Trimethylbenzene

To clarify the input of NMHC species, the following text has been added in the revised manuscript: "By taking the NMHC species incorporating in the MCM mechanism into account (MCM website, http://mcm.leeds.ac.uk/MCM/roots.htt, access date: 22 June 2020), observations of total 44 NMHC species, including 18 alkanes, 11 alkenes, ethyne and 14 aromatics were used as input for the model simulation (Table S2). The selected NMHCs contributed about 98% and 99% to the total mixing ratios and photochemical reactivities of all measured NMHCs at the Heshan site. Furthermore, the selected VOCs are the major precursors for Gly, Mgly, photochemical O3 and SOA (Ding et al., 2016, 2017; Li et al., 2014; Lou et al., 2010; Yuan et al., 2013), and have been frequently used to drive box model for studies on SOA, photochemical O3 and photochemical reactivity (Hofzumahaus et al., 2009; Lee-Taylor, et al., 2011)." For details, please refer to Lines 360-370, Page 13 in the revised manuscript. In addition, a table (Table S2) has been provided in the supplementary to list the input NMHCs.

2. The SOA simulation was not validated, thus it is not clear if the contribution of GLY and MGLY are in realistic range. For example, the study may underestimate or overestimate SOA from other species. Also, SOA formation pathways are far from accurate, if corrections such as wall loss are considered in this study? It is important to make sure that what recent findings regarding SOA formation have been considered, as the total SOA prediction would be much different. Reply: The reviewer's valuable comments are highly appreciated. We agreed with the reviewer that validation is important for SOA simulation although the aims of this study were to investigate the formation and sink of Gly and Mgly in the PRD region using an observation-based photochemical box model with the incorporation of improved mechanisms of heterogeneous processes of Gly

and Mgly, which would be further used to evaluate their contributions to SOA. As there were no direct SOA measurement data such as AMS data, we compared the model simulated SOA concentrations with those calculated by different methods and parameters, to evaluate the performance of SOA simulation. First, the EC (elemental carbon)-tracer method was used here to estimate the concentration of SOA in the present study, according to the equations 1-2 (Duan et al., 2005): SOC=ãĂŰOCãĂŮ_tot-EC×(OC∕EC)_min (1) SOA=S0C×ãĂŰCoefãĂŮ_(SOA/SOC) (2) where SOC represents the secondary organic carbon; OCtot represents the measured concentration of total organic carbon; (OC/EC)min is the emission ratio of primary OC (organic carbon) and EC (elemental carbon) from the sources, and can be represented by the minimum OC/EC ratio measured. Based on the (OC/EC)min value which was derived from the hourly observation data during the simulation period (i.e., January 07-08) using the minimum R squared (MRS) method and the hourly measured OC and EC concentrations (details in Chang et al., 2019), as well as the ratio of CoefSOA/SOC (Bae et al., 2006; Turpin and Lim, 2001), the mean concentration of SOA calculated by the EC-tracer method was 2.82 $\mu$g/m3, about 1.2 times the model simulated SOA concentration in the present study (2.47 $\mu$g/m3), suggesting that the PBM-MCM model provided a reasonable performance on the simulation of the magnitude of SOA, though uncertainties were frequently observed from the SOA concentrations calculated by the EC-tracer method. Second, we also conducted the comparison between measurement and simulation on other secondary products, including acetic acid, formic acid and pyruvic acid, which have been recognized as key SOA species, to further evaluate the model performance (Figure 1 as seen below). It was found that the simulated concentrations of acetic acid, formic acid and pyruvic acid were close to those observed at the Heshan site, accounting for ∼80%, 70% and 88% of observed values for acetic acid, formic acid and pyruvic acid, respectively. The results confirmed that secondary formation was the dominant source of above species at the Heshan site, and suggested that the PBM-MCM model could provide robust performance on simulating the abundance of above secondary species. The deviations between simulated and

observed concentrations may be related to the lack of consideration of primary emissions and/or other production pathways of above species in the model. Nevertheless, the above comparisons confirmed that the PBM-MCM model in this study indeed provided an appropriate description on the formation of SOA and other secondary organic products.

Fig. 1. The concentrations of acetic acid, formic acid and pyruvic acid in filter samples

In the model simulation, SOA formation is generally configured to occur through gas‐particle partitioning of low-volatility compounds formed by the gas‐phase oxidation of VOCs and other precursors (Aumont, et al., 2012; Lee-Taylor et al., 2011), which have been incorporated in the PBM-MCM model. On the other hand, the recent experimental results suggested that the formation of SOA in chambers may be suppressed due to the losses of SOA onto chamber walls, which leads to underestimation of SOA in air-quality and climate models (Matsunaga and Ziemann 2010; Zhang et al., 2014). We agreed with the reviewer that wall loss of SOA should be considered in the model configuration. Indeed, in this study, the wall loss of SOA has been considered in the model configuration, with the average wall loss rate coefficient of $6 \times 10-5$ s$-1$ according to the previous studies on the basis of the calculated organic material using an assumed density of 1 gÂůcm$-3$ (Johnson et al., 2004, 2005). In addition, the wall loss of other gaseous compounds (O3, NO2 and HNO3) were implemented in the box model with the average parameters of $3 \times 10-6$ s$-1$, $1.15 \times 10-5$ s$-1$ and $8.2 \times 10-5$ s$-1$, respectively. The detailed information for the calculation of above parameters was provided in Bloss et al. (2015). In addition, the results in the present study can reflect the current knowledge of abundance and evolution of Gly and Mgly, and the contributions of different VOCs to SOA formation in PRD to a certain extent. Since the study of Liggio et al. (2005), there have been many laboratory and model studies that explored and compared the heterogeneous uptake processes of Gly and Mgly on aqueous aerosols. For instance, the recent studies suggested that salting-in and salting-out effects (i.e., the ammonium-catalyzed reactions) had significant influences on the sur-

face uptake of dicarbonyls (Knote et al., 2014; Kampf et al., 2013; Noziere et al., 2009), and the rate coefficients were found to increase with the increasing ammonium ion activity (aNH4+) and pH (Noziere et al., 2009). Those new findings have led to more sophisticated descriptions of Gly and Mgly uptake on aerosols which should be considered accordingly in current model studies (Ervens et al., 2010, 2011). Therefore, in this study, we extended the PBM-MCM (version 3.2) model and included detailed gas-phase chemistry of Gly and Mgly formation, a module describing its partitioning and reactions in the aerosol aqueous-phase, as well as other updated heterogeneous processes, including the irreversible surface uptake process and reversible formation based on recent laboratory studies (Akimoto 2016; Houghton et al., 2017). Based on the above model development, we then compared the different pathways to form SOA from Gly, Mgly and other VOCs, respectively. In summary, according to the aims of this study mentioned above, our study indeed provided a comprehensive analysis on the abundance and evolution of Gly and Mgly in the PRD region with the incorporation of updated heterogeneous processes and gas-particle partitioning mechanisms, which were further used to investigate contributions of different VOCs to SOA formation. However, we admit that there were still some limitations in the model, e.g., the incorporation of more accurate and precise parameterization of heterogeneous process of Gly and Mgly (such as the surface uptake coefficients), the oxidation of other SOA precursors and complete mechanism of SOA formation, which needs further exploration in the future study (Sumner et al., 2014; Wu et al., 2019; Zhang and Seinfeld, 2013).

The brief discussion on the validation of model simulation has been added in the revised manuscript as follows, while the detailed information of validation has been provided in the supplementary (Section S3). "…….as there were no direct measured SOA data in this study (Chang et al., 2019), the model performance was evaluated by the comparison between the model simulated SOA with those calculated using the EC (elemental carbon)-tracer method, and by the comparison between the simulated and observed concentrations of other secondary products, which have been provided in detail in the supplementary (Section 3). For example, the simulated concentration of

SOA was about 85% of those calculated by the EC-tracer method based on the observed hourly data (Chang et al., 2019). Furthermore, the simulated concentrations of acetic acid, formic acid and pyruvic acid were close to those observed at the Heshan site, accounting for ∼80%, 70% and 88% of observed values for acetic acid, formic acid and pyruvic acid, respectively. The results confirmed that secondary formation was the dominant source of above species at the Heshan site, and suggested that the PBM-MCM model could provide robust performance on simulating the abundance of above secondary species and SOA." For details, please refer to Lines 404-417, Pages 14-15 in the revised manuscript and Section 3 in the supplementary.

In addition, to clarify the consideration of the influence of wall loss on SOA formation, the following text has been added in the revised manuscript: "The above gas-particle partitioning of low volatility compounds formed by the gas‐phase oxidation of VOCs and other precursors (Aumont, et al., 2012; Lee-Taylor et al., 2011) was configured in the model to estimate the SOA formation. However, the recent experimental results suggested that the formation of SOA in laboratory chambers may be suppressed due to losses of SOA to chamber walls, which leads to underestimates of SOA in air-quality and climate models (Matsunaga and Ziemann 2010; Zhang et al., 2014). Therefore, to consider the wall loss of SOA, the average wall loss rate coefficient of $6 \times 10-5$ s−1 was adopted in the model configuration according to previous studies on the basis of calculated organic material using an assumed density of 1 gÂůcm−3 (Johnson et al., 2004, 2005). In addition, the wall loss of other gaseous compounds (O3, NO2 and HNO3) were implemented in the box model with the average parameters of $3 \times 10-6$ s−1, $1.15 \times 10-5$ s−1 and $8.2 \times 10-5$ s−1, respectively. The detailed information for the calculation of above parameters was provided in Bloss et al. (2015)." For details, please refer to Lines 242-254, Pages 8-9 in the revised manuscript.

3. The basis for heterogeneous reactions are the comparison of gas phase of GLY and MGLY concentrations, which may cause uncertainties. Without validating the components formed by GLY and MGLY in SOA, it is misleading that GLY and MGLY are

actually converted to SOA. Reply: Thanks a lot for the reviewer's comment. We agree with the reviewer that there were possible uncertainties by only comparing the gas phase change of Gly and Mgly concentrations. As the measurement data of SOA and its speciation was not available in our field campaign, we could not directly compare the simulated and observed products formed by Gly and Mgly in SOA. As responded in above comment #2, the model simulated SOA concentration and some SOA components were compared with the measurement or measurement derived values, which show good consistency and suggest acceptable performance of SOA by the PBM-MCM model. Besides, by comparing with previous studies conducted in different regions in China, particularly those simulation studies of Gly/Mgly in the PRD region (Li et al., 2013, 2014), our study indeed provided a better reproduction of abundance and variations of Gly and Mgly at the Heshan site by adapting the gas-phase chemistry and the updated heterogeneous processes of Gly and Mgly (e.g., irreversible/reversible processes including the reversible partitioning, irreversible volume reactions, and surface uptake). Moreover, we investigated the difference in SOA production with and without the influence of Gly and Mgly in order to evaluate that whether that Gly and Mgly were actually converted to SOA. It was found that the SOA production has been improved apparently from 1.83 to 2.47 $\mu g$Åům-3 when considering the influence of Gly and Mgly. The total contribution of the Gly and Mgly contributed $\sim$26% to the simulated SOA concentration, of which $\sim$21% was from the heterogeneous processes of Gly and Mgly (SOAhet), further demonstrating that the heterogeneous processes have significant influences on the SOA formation of Gly and Mgly. Furthermore, it should be noted that gas phase Gly and Mgly could be removed by reactions with OH and NO3 radicals, photolysis and deposition (Sander et al., 2006; Volkamer., et al 2005). However, recent studies found that by comparing the removal of Gly and Mgly through OH, NO3 reactions, heterogeneous processes were the most important pathway for the destruction of Gly and Mgly (Shi, et al., 2020; De Haan, et al., 2018), and the lifetime of Gly and Mgly was reduced by a factor of two when the heterogeneous uptake processes were considered (Ervens and Volkamer 2010). This is consistent with the

results of the present study, which suggested that heterogeneous processes were the most important pathway for the destruction of Gly and Mgly (both with contributions of ∼62% during daytime), much higher than the removal of Gly and Mgly through other pathways. Therefore, by the improvement of SOA production with the consideration of Gly and Mgly and the good performance of the simulation of Gly, Mgly, SOA and other secondary products by the PBM-MCM model, it could be concluded that gaseous Gly and Mgly at Heshan were actually converted to SOA.

To highlight the contributions of Gly and Mgly to the SOA formation, the following text has been added in the manuscript: "The SOA production from Gly and Mgly was further explored by the model simulation with and without the consideration of Gly and Mgly. It was found that by incorporating the evolution of Gly and Mgly, the SOA production has been improved apparently from 1.83 to 2.47 $\mu$gÅům-3. The total contribution of the Gly and Mgly contributed ∼26% to the simulated SOA concentrations, of which ∼21% was from the heterogeneous processes of Gly and Mgly (SOAhet), further demonstrating that the heterogeneous processes have significant influences on the SOA formation from Gly and Mgly." For details, please refer to Lines 602-609, Pages 22 in the revised manuscript.

4. When considering the budget, it is import to considering transport. It is suggested to exclude the effects of transport in/out or note the readers regarding the uncertainties. Reply: Thanks a lot for the reviewer's comment. Due to the limitation of the PBM-MCM model and/or other box models (Aumont, et al., 2012; Lam et al., 2013; Lee-Taylor, et al., 2011), the influence of transport on air pollutants could not be estimated, though the simulation was conducted on the basis of observed mixing ratios of air pollutants that could be both influenced by local emissions and those transported from upwind areas (Liu et al.,2019).

To clarify the uncertainty associated with the lack of consideration of transport in the model, the following text has been added: "Similar to other box models in simulating the degradation of VOCs and formation of SOA (Aumont, et al., 2011; Lee-Taylor, et al.,

[revised manuscript text omitted]

reaction of glyoxal catalyzed by ammonium ions (NH4+). J. Phys. Chem. A. 113(1), 231-237. Sander, S. P., Golden, D. M., Kurylo, M. J., Moortgat, G. K., Wine, P. H., Ravishankara, A. R., Orkin, V. L., 2006. Chemical kinetics and photochemical data for use in atmospheric studies evaluation number 15. Pasadena, CA: Jet Propulsion Laboratory. National Aeronautics and Space Administration. 2006, 97–4(2000):1135-1151. Shi, Q., Zhang, W., Ji, Y., Wang, J., Qin, D., Chen, J., An, T., 2020. Enhanced uptake of glyoxal at the acidic nanoparticle interface: implications for secondary organic aerosol formation. Environmental Science: Nano, 7(4), 1126-1135. Sumner, A. J., Woo, J. L., McNeill, V. F., 2014. Model Analysis of secondary organic aerosol formation by glyoxal in laboratory studies: The case for photoenhanced chemistry. Environ. Sci. Technol. 4820, 11919-11925. Turpin, B. J., Lim, H. J., 2001. Species contributions to PM2. 5 mass concentrations: Revisiting common assumptions for estimating organic mass. Aerosol Sci Technol. 35(1), 602-610. Volkamer, R., Molina, L.T., Molina, M.J., Shirley, T. and Brune, W.H., 2005. DOAS measurement of glyoxal as an indicator for fast VOC chemistry in urban air. Geophys. Res. Lett. 32(8), L08806. Waxman, E. M., Elm, J., Kurtén, T., Mikkelsen, K. V., Ziemann, P. J., Volkamer, R., 2015. Glyoxal and methylglyoxal setschenow salting constants in sulfate, nitrate, and chloride solutions: Measurements and Gibbs energies. Environ. Sci. Technol. 4919, 11500-11508. Wu, L.Q., Wang, X.M., Lu, S.H., Shao, M., Ling, Z.H., 2019. Emission inventory of semi-volatile and intermediate-volatility organic compounds and their effects on secondary organic aerosol over the Pearl River Delta region. . Atmos. Chem. Phys. 19, 8141-8161. Yuan, B., Hu, W.W., Shao, M., Wang, M., Chen, W.T., Lu, S.H., Zeng, L.M., Hu, M., 2013. VOC emissions, evolutions and contributions to SOA formation at a receptor site in eastern China. Atmos. Chem. Phys. 13, 8815-8832. Yun, H., Wang, W., Wang, T., Xia, M., Yu, C., Wang, Z., Poon, S. C. N., Yue, D., Zhou, Y., 2018. Nitrate formation from heterogeneous uptake of dinitrogen pentoxide during a severe winter haze in southern China. Atmos. Chem. Phys. 18(23), 17515-17527. Zhang, X. and Seinfeld, J.H., 2013. A functional group oxidation model (FGOM) for SOA formation and aging. Atmos. Chem. Phys. 13, 5907-5926. Zhang, X., Cappa,

C. D., Jathar, S. H., McVay, R. C., Ensberg, J. J., Kleeman, M. J., Seinfeld, J. H. 2014. Influence of vapor wall loss in laboratory chambers on yields of secondary organic aerosol. Proceedings of the National Academy of Sciences. 111(16), 5802-5807.

Please also note the supplement to this comment:
https://www.atmos-chem-phys-discuss.net/acp-2020-212/acp-2020-212-AC1-supplement.pdf

---

## Author Comment (AC2) · 21 Jul 2020

Zhenhao Ling et al. lingzhh3@mail.sysu.edu.cn

Response to Reviewers

Reviewer #2 The paper investigates the importance of glyoxal (GLY) and methylglyoxal (MGLY) on SOA formation in the PRD region. The importance of GLY and MGLY on the

[Figure]

SOA formation has been previously studied but this study investigated several different methods of modeling GLY and MGLY SOA, which provides additional information on how to model this important process. I have a few suggestions for the authors when they revise their paper: Reply: Thank a lot for the reviewer's positive comments and helpful suggestions. We have addressed all the comments/suggestions in the revised manuscript. Detailed responses to the individual specific comment/suggestion are as follows. All the revision is highlighted in the revised manuscript.

1. The gas-phase concentrations of OH/HO2 are not constrained in the box model simulations. Instead, they are calculated using the box model. However, GLY and MGLY can be removed in the gas phase reactions with OH. Thus, it is necessary to evaluate the predicted OH/HO2 concentrations to make sure the competing between gas/particle partitioning, which forms SOA, and the gas phase decay processes that reduce the SOA formation is correctly captured. Reply: The reviewer's comment is highly appreciated. In this study, the OH and HO2 radicals were simulated based on the measured mixing ratios of O3, NOx, CO, and NMHCs, together with meteorological parameters, as the direct measurement of OH and HO2 radical was not available (Xue et al., 2014a, b). Furthermore, the photolysis rates adopted in the model were determined from the photon fluxes from the Tropospheric Ultraviolet and Visible Radiation (TUV-v5) model (Madronich and Flocke 1997) according to the sampling location and modeling period. Previous studies have demonstrated that the observation-based model coupled with Master Chemical Mechanism (MCM), such as the PBM-MCM model in the present study, could perform well in the simulation of O3, photochemical reactivities and atmospheric radical chemistry (e.g., the budgets, variations of OH and HO2 radicals and contributions from varied production and destruction pathways) in different environments/regions in China (Huang et al., 2020 and references therein; Ling et al., 2014; Liu et al., 2019; Wang et al., 2018; Xue et al., 2014a, b; Yang et al., 2018). To better evaluate the model performance on the simulation of OH and HO2 abundance and variations, we also compared our simulation results with the previous observations conducted in PRD and other regions in the world (Table 1 as seen below). In this

study, the simulation on the diurnal variations of OH and HO2 was performed well, with peak values at noon, consistent with those measured and simulated in PRD (Hofzuma-haus et al., 2009 and related papers; Tan et al., 2019). The simulated mean mixing ratios of OH and HO2 radicals from the model in the present study were ∼1.6 ×106 moleculeÂůcm-3 and ∼3 ×107 moleculeÂůcm-3, which are comparable to the winter observations at Beijing, Tokyo, and New York (Kanaya et al., 2007; Ren et al., 2006; Ma et al., 2019), and lower than the measurement and simulation values in summer (e.g., July) or autumn (e.g., October to November) in the PRD region (Table S3 in the sup-plementary) (Hofzumahaus et al., 2009; Tan et al., 2019). Note that the variations of simulation results in the present study and those observation results in previous stud-ies in PRD may be associated with differences in the levels of O3 and its precursors, different photolysis rates, and to a lesser extent, meteorological conditions (Hofzuma-haus et al., 2009). The higher OH and HO2 mixing ratios were expected in summer and autumn than winter due to the stronger solar radiation and higher temperature, as well as the variations of O3 and its precursors in different sites, though the mea-surement of OH/HO2 radicals has been very challenging, and significant uncertainties still exist in the measurement values of the radicals (Hofzumahaus et al., 2009; Tan et al., 2019). Furthermore, the comparison between the simulation of a box model and observation results suggested that the higher observed mixing ratios of OH and HO2 radicals were related to an unidentified source of OH at the backgarden site of PRD in summer of 2006, while the comparison between the observed OH/HO2 variations and those calculated from the parameterization of HOx (HOx = OH + HO2) produc-tion and destruction indicated a missing OH source of 4-6 ppbvÂůh-1 and an unknown RO2 loss at the Heshan site in autumn of 2014. Overall, the above comparison sug-gested that the model simulation for the variations of OH and HO2 radicals, and related atmospheric photochemical reactivities was appropriate at Heshan in this study.

Table 1 The mean measured OH and HO2 concentrations in the previous studies and our model results (in molecule cm-3) Location season OH (×106) HO2 (×108) year Reference PRD, China Summer 15 - 2006 (Hofzumahaus et al., 2009) Shanghai, China

10.2 - 2013 (Nan et al., 2017) north-western Greece 8 4.7 1997 (Creasey et al., 2001) PRD, China Autumn 4.5 3.0 2014 (Tan et al., 2019) Tokyo, Japan Winter 1.5 - 2004 (Kanaya et al., 2007)

New York, America 1.4 - 2004 (Ren et al., 2006)

Beijing, China 1.5 0.3 2017 (Ma et al., 2019)

PRD, China 1.6 0.3 2017 This study

To clarify the model performance on the simulation of OH and HO2 abundance and variations, the above analysis and comparison have been added in the revised manuscript: "In this study, the simulation on the diurnal variations of OH and HO2 was performed well, with peak values at noon, consistent with those measured and simulated in PRD (Hofzumahaus et al., 2009 and related papers; Tan et al., 2019). The simulated mean mixing ratios of OH and HO2 radicals from the model in the present study were ∼1.6 ×106 moleculeÂůcm-3 and ∼3 ×107 moleculeÂůcm-3, which are comparable to the winter observations at Beijing, Tokyo, and New York (Kanaya et al., 2007; Ren et al., 2006; Ma et al., 2019), and lower than the measurement and simulation values in summer (e.g., July) or autumn (e.g., October to November) in the PRD region (Table S3 in the supplementary) (Hofzumahaus et al., 2009; Tan et al., 2019). Note that the variations of simulation results in the present study and those observation results in previous studies in PRD may be associated with differences in the levels of O3 and its precursors, different photolysis rates, and to a lesser extent, meteorological conditions (Hofzumahaus et al., 2009). The higher OH and HO2 mixing ratios were expected in summer and autumn than winter due to the stronger solar radiation and higher temperature, as well as the variations of O3 and its precursors in different sites, though the measurement of OH/HO2 radicals has been very challenging, and significant uncertainties still exist in the measurement values of the radicals (Hofzumahaus et al., 2009; Tan et al., 2019). Furthermore, the comparison between the simulation of a box model and observation results suggested that the higher observed mixing ratios of

OH and HO2 radicals were related to an unidentified source of OH at the backgarden site of PRD in summer of 2006, while the comparison between the observed OH/HO2 variations and those calculated from the parameterization of HOx (HOx = OH + HO2) production and destruction indicated a missing OH source of 4-6 ppbvÂůh-1 and an unknown RO2 loss at the Heshan site in autumn of 2014." For details, please refer to Lines 380-403, Page 14 in the revised manuscript and Table S3 in the supplementary.

2. The impact of O3 on GLY and MGLY is not discussed. Looking at Figure 2, it is obvious that GLY and MGLY must decrease at night. GLY and MGLY can also react with O3. This is likely an import process that reduces GLY and MGLY at night, in addition to SOA formation from gas-to-particle partitioning and aqueous reactions. Since GLY and MGLY data are collected throughout the entire 24 hour period, it might be interesting to see how well the box model predicts GLY and MGLY at night with different SOA modeling approaches. The nighttime behavior of GLY and MGLY and their roles in SOA formation is not as clear as the daytime and should not be ignored in this study. Reply: Thanks for the reviewer's comment. By investigating the relative contributions of different loss pathways of Gly and Mgly, it was found that during the daytime, the heterogeneous processes were the most important pathway for the destruction of Gly and Mgly (both with contributions of ∼62%), followed by photolysis, OH reactions and dry deposition. We also examined the loss pathways of Gly and Mglys during nighttime, and only the heterogeneous processes make notable contribution to Gly and Mgly destruction, accounting for more than 90% of the total destruction (Table 2 as seen below), which was consistent with previous studies (Washenfelder et al., 2011; Gomez et al., 2015). The lower contributions of Gly and Mgly with radicals were mainly because of the low concentration of OH at night and their relatively lower reactivities with NO3 radical (Calvert et al., 2011; Mellouki et al., 2015). Table 2. The relative contributions of different loss pathways of Gly and Mgly at nighttime Loss pathyways Gly Mgly NO3,OH-reaction, % 2.44 3.56 dry deposition, % 1.04 0.74 dilution, % 1.92 1.60 heterogeneous a, % Irreversible processes, % 57.9 55.9 Reversible processes, % 36.7 38.2 a Considered both irreversible and reversible parameterizations

of the aerosol sinks (i.e., scenario 4 and M2).

Furthermore, we agree with the reviewer that Gly and Mgly may be also removed by the reaction with O3, and incorporation of more possible reaction mechanisms in addition to oxidation of Gly and Mgly by NO3 and OH radicals are reasonable. The previous study reported that kinetic data of O3 reactions with Gly and Mgly are of negligible atmospheric importance, with the reaction rate constants of < 3 and < 6 × 10-21 cm3Åůmolecule-1Åůs-1, respectively, which are 6 order of magnitude lower than the reaction rate constants with NO3 (which >1 and > 2 × 10-15 cm3Åůmolecule-1Åůs-1, respectively), and are 9 order of magnitude lower than the reaction rate constants with OH (which = 9 and 13 × 10-12 cm3Åůmolecule-1Åůs-1) (Mellouki et al., 2015). Due to the much lower reaction rates, we believe that the influence of O3 on the removal of Gly and Mgly was negligible. Furthermore, there were few parameterizations for the reaction mechanism of Gly/Mgly with O3 due to their low reaction rates with O3. Therefore, the present study did not include the pathway of O3 oxidation on Gly and Mgly. To explain the exclusion of the oxidation of Gly and Mgly by O3, and to clarify the contributions of different pathways to the removal of Gly and Mgly during nighttime, the following text has been added in the revised manuscript: "It should be noted that the oxidation of Gly and Mgly by O3 was not considered in this study as the reaction rate constants of Gly and Mgly with O3 are < 3 and < 6 × 10-21 cm3Åůmolecule-1Åůs-1, respectively, which are 6 order of magnitude lower than the reaction rate constants with NO3 (which >1 and > 2 × 10-15 cm3Åůmolecule-1Åůs-1, respectively), and are 9 order of magnitude lower than the reaction rate constants with OH (9 and 13 × 10-12 cm3Åůmolecule-1Åůs-1 for the reactions of Gly and Mgly with O3, respectively) (Mellouki et al., 2015). Therefore, we believe that the influence of O3 on the removal of Gly and Mgly was negligible (Mellouki et al., 2015). Furthermore, there were few parameterizations for the reaction mechanism of Gly/Mgly with O3 due to their low reaction rates with O3. On the other hand, at nighttime, only the heterogenous processes made the main contribution to Gly and Mgly destruction, with contributions higher than 90% to the total destruction of Gly and Mgly at night (Table S9 in the supplementary), consis-

tent with previous studies (Washenfelder et al., 2011; Gomez et al., 2015). The lower contributions of Gly and Mgly with radicals were mainly because of the low OH concentration at night and their relatively lower reactivities with NO3 radical (e.g., the reaction rate constants of Gly/Mgly with NO3 are ∼1000 times lower than those with OH radical) (Calvert et al., 2011; Mellouki et al., 2015)." For details, please refer to Lines 544-562, Pages 19-20 in the revised manuscript and Table S9 in the supplementary.

3. The other issue that I think should be addressed is the primary emissions of GLY and MGLY, since not all are produced secondarily. It seems that no emissions of primary GLY and MGLY are included in the box model simulations. The authors might want to discuss how this omission can impact their conclusions. Reply: We thank the reviewer's valuable comment. Indeed, not all the Gly and Mgly in the atmosphere are produced secondarily. However, many previous studies have suggested that the dicarbonyls such as Gly and Mgly have limited primary sources except biomass burning and biofuel combustion (Grosjean et al., 2001; Zhang et al., 2016). The primary emissions of Gly and Mgly were much less significant than those secondarily from photochemical reactions (Lv et al., 2019). Fu et al. (2008) estimated that the primary emissions only accounted for about 4% and 17% to the total emissions of Mgly and Gly, respectively. To preliminarily estimate the contributions of primary and secondary sources to measured Gly and Mgly, a correlation-based source apportionment method suggested by previous studies was used, as described by equation 1 (Friedfeld et al., 2002; Yuan et al., 2013) . $[\text{dicarbonyls}] = \beta\_0 + \beta\_1 [C\_2 H\_2] + \beta\_2 [O\_3]$ (1) where $\beta0$, $\beta1$, and $\beta2$ are the coefficients derived from the linear regression analysis. For every unit increase in C2H2 concentration there is a $\beta1$ unit increase in dicarbonyls. Similarly, for every unit increase in O3 concentration there is a $\beta2$ unit increase in dicarbonyls. $\beta\_0$ can be considered the background carbonyls level (in units of ppbv). Relative contributions from primary emissions, secondary formation, and background dicarbonyl concentrations can be computed according to the tracer concentrations and corresponding $\beta$-values by the following equations: $P\_{primary} = (\beta\_1 [C\_2 H\_2]i) / ((\beta\_0 + \beta\_1 [C\_2 H\_2]i + \beta\_2 [O\_3]i)) \times 100\%$ (2) $P\_{secondary} = (\beta\_2 [O\_3]i) / ((\beta\_0 + \beta\_1 [C\_2 H\_2]i + \beta\_2$

[O_3 ]i) )×100% (3) P_background=$\beta$_0/(($\beta$_0+$\beta$_1 [C_2 H_2 ]i+$\beta$_2 [O_3 ]i) )×100% (4) Table 3 shows linear regression coefficients and relative source contributions of Gly and Mgly. It was found that the contributions from primary sources is significantly lower than those from secondary sources (96.14% and 96.44% for Gly and Mgly, respectively), confirming that Gly and Mgly in the present study were mostly related to secondary formation.

Table 3. Linear regression coefficients and relative contributions of primary, secondary, and background sources of Gly and Mgly. Linear regression coefficients Source Contribution $\beta$_0 $\beta$_1 $\beta$_2 Background Primary Secondary R Gly 0.001 0.066 0.042 0.40% 3.46% 96.14% 0.77 Mgly 0.002 0.108 0.081 0.05% 3.51% 96.44% 0.75

To highlight the potential significance of secondary formation on the abundance of Gly and Mgly, the following text was added: "It has been well documented that Gly and Mgly have limited primary sources except biomass burning and biofuel combustion (Grosjean et al., 2001; Zhang et al., 2016). Furthermore, the primary emissions of Gly and Mgly were much less significant than those secondarily from photochemical reactions (Lv et al., 2019). Fu et al. (2008) estimated that primary emissions only accounted for about 4% and 17% to the total emissions of Mgly and Gly, respectively." For details, please refer to Lines 80-85, Page 3 in the revised manuscript.

Furthermore, the following text for the preliminary estimation of primary and secondary sources of Gly and Mgly was added in the revised manuscript: "In this study, the simulated Gly and Mgly were secondarily formed from the oxidation of their VOC precursors. Therefore, before the comparison between the simulation and observation results, the contributions of primary and secondary sources to the measured Gly and Mgly were preliminarily estimated by a correlation-based source apportionment method suggested by previous studies (Friedfeld et al., 2002; Yuan et al., 2013). Table S5 in the supplementary shows linear regression coefficients and relative source contributions of Gly and Mgly. It was found that the contributions from primary sources (3.46 % and 3.51% for Gly and Mgly, respectively) were significantly lower than those

from secondary sources (96.14% and 96.44%, respectively), confirming that observed Gly and Mgly in the present study were mostly related to secondary formation." For details, please refer to Lines 440-450, Page 16 in the revised manuscript and Table S5 in the supplementary.

4. Modeling SOA formation from GLY and MGLY using the Master Chemical Mechanism has been discussed in a more complete regional air quality model that considers the transport and emissions of GLY and MGLY(Li et al. 2015). Another paper that discusses the importance of the GLY and MGLY on SOA formation is Ying et al 2015, which shows that including SOA from uptake of GLY/MGLY leads to significant improvement in predicated SOA in these southeast United Stated. These references appear to be neglected by the authors. Reply: Thanks for the reviewer's suggestions. It has been revised accordingly in the revised manuscript. "For example, Li et al (2015) constructed a Master Chemical Mechanism with an equilibrium partitioning module and coupled it in a Community Air Quality Model (CMAQ) to predict the regional concentrations of SOA from VOCs in the eastern United States (U.S). It was found that those SOA formed from Gly and Mgly were accounted for more than 35% of total SOA. Similarly, Ying et al. (2015) used a modified SAPRC-11 (S11) photochemical mechanism, considering the surface-controlled reactive uptake of Gly and Mgly, and incorporated the mechanism in the CMAQ model to simulate ambient SOA concentrations during summer in the eastern U.S. The results showed that the uptake of Gly and Mgly resulted in the significant improvement in predicated SOA concentration, and the aerosol surface uptake of isoprene-generated Gly, Mgly and epoxydiol accounted for more than 45% of total SOA." For details, please refer to Lines 68-78, Page 3 in the revised manuscript.

References: Calvert, J., Mellouki, A., Orlando, J., 2011. Mechanisms of atmospheric oxidation of the oxygenates. OUP USA. Creasey, D.J., Heard, D.E., Lee, J.D., 2001. OH and HO2 measurements in a forested region of north-western Greece. Atmos. Environ. 35(27), 4713-4724. Friedfeld, S., Fraser, M., Ensor, K., Tribble, S., Rehle, D., Leleux, D. and Tittel, F., 2002. Statistical analysis of primary and secondary atmospheric formaldehyde. Atmos. Environ. 36(30), 4767-4775. Fu, T.-M., Jacob, D.J., Wittrock, F., Burrows, J.P., Henze, D.K., 2008. Global budgets of atmospheric glyoxal and methylglyoxal, and implications for formation of secondary organic aerosols. J. Geophys. Res. Atmos. 113(D15). Fuchs, H., Hofzumahaus, A., Rohrer, F., Bohn, B., Brauers, T., Dorn, H. P., Haseler, R., Holland, F., Kaminski, M., Li, X.,Lu, K., Nehr, S., Tillmann, R., Wegener, R., and Wahner, A., 2013. Experimental evidence for efficient hydroxyl radical regeneration in isoprene oxidation, Nat. Geosci., 6, 1023–1026. Garcia, AR, Volkamer, R, Molina, LT, Molina, MJ, Samuelson, J, Mellqvist, J, Galle, B, Herndon, SC, Kolb, CE. 2005. Separation of emitted and photochemical formaldehyde in Mexico City using a statistical analysis and a new pair of gas-phase tracers. Atmos. Chem. Phys Discussions, European Geosciences Union, 2005, 5 (6),11583-11615. Gomez, M.E., Lin, Y., Guo, S., Zhang, R., 2015. Heterogeneous chemistry of glyoxal on acidic solutions. An oligomerization pathway for secondary organic aerosol formation. J. Phys. Chem. A. 119(19). 4457-4463. Grosjean, D, Grosjean, E, Gertler, A., 2001. On-road emissions of carbonyls from light-duty and heavy-duty vehicles. Environ. Sci. Technol. 351, 45-53. Hofzumahaus, A., Rohrer, F., Lu, K.D., Bohn, B., Brauers, T., Chang, C.C., Fuchs, H., Holland, F., Kita, K., Kondo, Y., Li, X., Lou, S.R., Shao, M., Zeng, L., Wahner, A., Zhang, Y.H., 2009. Amplified trace gas removal in the troposphere. Science. 324, 1702-1704. Huang, W.W., Zhao, Q.Y., Liu, Q., Chen, F., He, Z.R., Guo, H., Ling, Z.H., 2020. Assessment of atmospheric photochemical reactivity in the Yangtze River Delta using a photochemical box model. Atmos. Res. 245, 105088. Kanaya, Y., Cao, R., Akimoto, H., Fukuda, M., Komazaki, Y., Yokouchi, Y., Koike, M., Tanimoto, H., Takegawa, N., Kondo, Y., 2007. Urban photochemistry in central Tokyo: 1. Observed and modeled OH and HO2 radical concentrations during the winter and summer of 2004. J. Geophys. Res. Atmos. 112(D21). Li, J., Cleveland, M., Ziemba, L. D., Griffin, R. J., Barsanti, K. C., Pankow, J. F., 2015. Modeling regional secondary organic aerosol using the Master Chemical Mechanism. Atmos. Environ. 102, 52-61. Li, X., Rohrer, F, Brauers, T, Hofzumahaus, A, Lu, K, Shao, M, Zhang, YH, Wahner, A. 2014. Modeling of HCHO and CHOCHO at a semi-rural

[revised manuscript text omitted]

Please also note the supplement to this comment:

https://www.atmos-chem-phys-discuss.net/acp-2020-212/acp-2020-212-AC2-supplement.pdf
* * *